# Gauging permutation symmetries
# as a route to non-Abelian fractons

**Abhinav Prem[1⋆] and Dominic J. Williamson[2]**

**1** Princeton Center for Theoretical Science, Princeton University, NJ 08544, USA
**2** Department of Physics, Yale University, New Haven, CT 06520-8120, USA

Both authors contributed equally to this work

## Abstract

We discuss the procedure for gauging on-site $\mathbb{Z}_2$ global symmetries of three-dimensional lattice Hamiltonians that permute quasi-particles and provide general arguments demonstrating the non-Abelian character of the resultant gauged theories. We then apply this general procedure to lattice models of several well known fracton phases: two copies of the X-Cube model, two copies of Haah's cubic code, and the checkerboard model. Where the former two models possess an on-site $\mathbb{Z}_2$ layer exchange symmetry, that of the latter is generated by the Hadamard gate. For each of these models, upon gauging, we find non-Abelian subdimensional excitations, including *non-Abelian fractons*, as well as non-Abelian looplike excitations and Abelian fully mobile pointlike excitations. By showing that the looplike excitations braid non-trivially with the subdimensional excitations, we thus discover a novel gapped quantum order in 3D, which we term a *"panoptic" fracton order*. This points to the existence of parent states in 3D from which both topological quantum field theories and fracton states may descend via quasi-particle condensation. The gauged cubic code model represents the first example of a gapped 3D phase supporting (inextricably) non-Abelian fractons that are created at the corners of fractal operators.

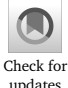

## Contents



# 1   Introduction

A new chapter in the book of three-dimensional (3D) quantum phases of matter was opened with the discovery of fracton models [1–8], characterized by the presence of topological excitations with restricted mobility. These peculiar particles have attracted significant recent interest, thereby revealing intriguing connections to quantum information processing [9–12], topological order [13–22], sub-system symmetries [23–30], and slow quantum dynamics [1,3,31–33]. Much of the phenomenology of fractons can also be realized in tensor gauge theories [34–44] with higher moment conservation laws, unveiling further connections of fractons with elasticity [45–49] and even gravity [50–52]. For a recent review of fractonic physics, we refer the reader to Ref. [53].

A large class of *gapped* fracton models are described by local commuting projector Hamiltonians, thereby enabling their exact, analytic study. Broadly, these are classified into type-I or type-II models, with the latter distinguished by the lack of *any* string-operators and therefore also any topologically non-trivial mobile quasi-particles. Similar to conventional topologically ordered states, gapped fracton phases possess long-range entangled ground states [29,54–58], the number of which grows sub-extensively with system size on topologically non-trivial manifolds. The dependence of the ground state degeneracy (GSD) upon the system size is a manifestation of the geometric sensitivity of fracton order [51,52,59–65], rendering these phases "beyond" the familiar topological quantum field theory (TQFT) paradigm for liquid topological orders. Thus, despite the rapid progress in characterizing fracton order, a unified mathematical framework describing it has so far proved elusive.

This is in contrast with 2D topological orders, whose classification in terms of unitary modular tensor categories [66] (see e.g., Ref. [67]) was facilitated by exactly solvable lattice models, in both two and three spatial dimensions [68–72]. Two paradigmatic classes of such exactly soluble models are Kitaev's quantum double models [73] and the Levin-Wen string-net models [74], which provided important insights about the nature of topological order. Taken together, these models encapsulate the key features of 2D long-range entangled phases (in the absence of global symmetries) and provide a general framework within which fractionalized excitations, including those with *non-Abelian* character, are realized. A key step towards categorizing fracton order is thus developing classes of exactly solvable models which can capture a wide range of fracton phases, including those with non-Abelian excitations. Such excitations are not precluded in fracton phases despite general arguments restricting the statistics of pointlike excitations in 3D, since these arguments assume complete particle

mobility. The subdimensional nature of excitations[1] in fracton phases thus opens the door to non-Abelian pointlike particles with nontrivial braiding statistics in three dimensions; hence, besides providing fundamental insights into the nature of 3D gapped quantum phases, finding non-Abelian fracton orders could have potential implications for quantum information storage and processing.

There has been recent progress on this front [62, 75, 76]: in Ref. [62], layers of 2D string-nets were coupled to produce fracton phases with non-Abelian excitations mobile only along lines, while non-Abelian fractons were found by twisting the layers of Abelian fracton models in Ref. [76]. Although they provide a large class of novel non-Abelian fracton models, it does not appear that the cage-net and twisted fracton models exhaust all possible fracton orders. Ref. [63], for instance, coupled layers of 2D Abelian topological orders to a 3D Abelian topological phase to reproduce the X-Cube model [8] and generalizations thereof—we anticipate that suitable non-Abelian generalizations of the "string-membrane-net" can produce fracton orders distinct from those found in the cage-net or twisted fracton models. Moreover, all of the aforementioned examples fall into the category of *foliated* type-I fracton phases, where the presence of excitations mobile along lines or planes allows generalized notions of braiding and statistics to be defined [19, 21]. Unlike these examples, where notions of "non-Abelianness"[2] have been extended to subdimensional excitations [62, 76], whether such notions extend to type-II models, lacking any mobile excitations, remains unclear. An obvious obstruction to studying this question is the lack of *any* non-Abelian type-II model in the literature.

In this paper, we add another wrinkle to the developing story of 3D gapped phases by finding new exactly solvable models which *simultaneously* host non-Abelian subdimensional particles, including non-Abelian fractons, and non-Abelian looplike excitations, with non-trivial braiding betwixt these distinct sectors. Our starting point is the observation that gauging an on-site global symmetry that acts as layer swap[3] on copies of (2D or 3D) topological orders leads to a non-Abelian topological order[4]—such symmetries fall under the umbrella of more general "anyonic" or "anyon permutation" symmetries, which, when gauged, can enlarge the topological order of the underlying symmetry enriched phase [79–81]. In principle, gauging such on-site global symmetries is a well defined procedure which can be carried out for any many-body quantum state invariant under that symmetry. Gauging a global on-site symmetry which exchanges subdimensional excitations in some fracton order thus proffers a natural route for obtaining non-Abelian subdimensional particles, including fully immobile non-Abelian fractons. Physically, the gauging procedure corresponds to condensing (or melting) symmetry defects which occur at the boundaries of domain walls in the symmetry-enriched topological order.

We discuss the general gauging procedure here, which when applied to copies of some exactly solvable gapped Hamiltonian with a global SWAP (or Hadamard) symmetry, produces another exactly solvable gapped Hamiltonian. Aside from providing a constructive framework for producing solvable models describing the gauged phase, we are also able to obtain important general results regarding the nature of the gauged Hamiltonian. In particular, we show that gauging a $\mathbb{Z}_2$ swap symmetry between layers of *any* 3D topological phase, including fracton orders, with some pointlike particles results in non-Abelian deconfined excitations in the gauged phase. Additionally, the symmetry defects (also called "genons" [82]) in the symmetry-enriched phase proliferate upon gauging, so that the gauged phase hosts non-Abelian looplike

---

[1]Here, by subdimensional we refer to excitations that are fully immobile (fractons), mobile only along lines (lineons), or mobile only along planes (planons) of the 3D lattice.

[2]Non-Abelian excitations are those which participate in multiple fusion channels and can thus encode quantum information non-locally throughout the system. See e.g., Refs. [77, 78].

[3]We interchangeably refer to this global symmetry as swap, layer-swap, or layer exchange symmetry.

[4]This gauging technique, when applied to two copies of a quantum double $\mathcal{D}(G)$, leads to a model in the same phase as $\mathcal{D}(\tilde{G})$, with $\tilde{G} = (G \times G) \rtimes \mathbb{Z}_2$ which is non-Abelian even when the underlying group $G$ is Abelian.

excitations, which are the gauge flux loops. We also provide a general argument that guarantees the absence of stringlike operators creating pairs of fractons in the gauged theory as long as such operators do not exist in the underlying symmetry-enriched phase. In other words, gauging a fracton permuting symmetry *necessarily* produces deconfined non-Abelian fractons, alongside other Abelian particles and non-Abelian loop excitations.

Our arguments further demonstrate that the gauged phase hosts an *entirely new* quantum order, since the subdimensional excitations braid non-trivially with the gauge flux loops—this statistical interaction ensures that the phase is not local unitary equivalent to some non-Abelian fracton model decoupled from some non-Abelian 3D topological order. Instead, the phases obtained by gauging $\mathbb{Z}_2$ layer exchange symmetries support what we dub a *panoptic fracton order*, wherein subdimensional excitations coexist with, and braid non-trivially with, looplike excitations. As such, these phases present a hitherto undiscovered generalization of fracton order, most models for which have heretofore consisted *only* of subdimensional particles—the only exceptions are the type-I string-membrane-net models presented in Ref. [63], which also contained fractons, loops, and fully mobile particles. However, the construction here presents a larger class as it can account for non-Abelian fractons even in type-II models. We also stress that while the gauged phases contain looplike excitations, suggestive of 3D topological order, they retain the characteristic geometric sensitivity of fracton phases, thereby remaining outside the usual TQFT paradigm.

While our general arguments are sufficient for conceptually establishing the existence of a broader, more inclusive fracton order, we explicitly apply the gauging map to familiar fracton models: the X-Cube model [8], the checkerboard model [7], and Haah's cubic code [2]. For each of these, we provide the explicit form of the gauged Hamiltonian as well as the relevant Wilson string or membrane operators. We also discuss the effect of gauging in terms of the quasi-particles of the type-I models and show the existence of *inextricably* non-Abelian fractons [62, 76], which shows their fundamentally three-dimensional nature. When applied to Haah's cubic code, our general formalism produces, to the best of our knowledge, the first model where inextricably non-Abelian fractons are created at the corners of a *fractal* operator. Since the gauging map also produces non-Abelian gauge flux loops, there is a well-defined statistical phase associated with braiding the loop around an isolated fracton, which allows this model to evade the obstacle of defining non-Abelianness in models with *only* immobile fractons. The gauged model is, strictly speaking, no longer of type-II as there appear additional *fully mobile* Abelian pointlike particles, and instead falls into the broader category of panoptic fracton phases.

The rest of the paper is organized as follows: we describe the general procedure for gauging on-site local symmetries of many-body states in Sec. 2, providing both a lattice description and a quasi-particle description. Focusing on global $\mathbb{Z}_2$ symmetries that act as layer swap on two copies of some topological (possibly fracton) order, we develop a general argument in Sec. 2.3 demonstrating that the gauging procedure generates non-Abelian particles and looplike excitations. We further show that the mobility restrictions in a symmetry-enriched fracton phase are retained in the gauged phase, such that no string-like operators can move the gauged fractons. We apply the general gauging procedure to well-known fracton models in Sec. 3, focusing on the X-cube, the checkerboard, and the cubic code models. For each of these, we provide the explicit gauged Hamiltonian alongside a description of the emergent quasi-particles, of which several are non-Abelian. We conclude with a discussion of open questions and future directions. For completeness and to illustrate the significant qualitative differences between fracton and topological orders, the gauging procedure is applied to the 2D and 3D toric codes, as well as the 2D color code, in the Appendices.

# 2 Gauging a $\mathbb{Z}_2$ symmetry

## 2.1 Lattice description

We start with a brief discussion of gauging the on-site global symmetry of a many-body quantum state. In the usual gauging procedure, which operates at the level of a Lagrangian or Hamiltonian description of the matter fields, operators of the globally symmetric theory are modified through the minimal coupling procedure into gauged operators. Here, however, we describe how gauging proceeds at the level of many-body quantum states directly, independent of any prescribed dynamics for the matter fields. For simplicity, we restrict our attention to $\mathbb{Z}_2$ global symmetries and refer the reader to Refs. [83,84] for the general case, which includes non-Abelian symmetries.

Consider a lattice (more generally, a graph) $\Lambda$ with matter degrees of freedom living on its vertices. Each vertex is thus endowed with a Hilbert space $\mathbb{H}_v$, such that the total matter Hilbert space $\mathbb{H}^M := \otimes_{v \in \Lambda} \mathbb{H}_v$. The edges of the lattice are denoted $e \in \Lambda$ and can be left unoriented when considering Abelian global symmetries.

Let $U_v(g)$ correspond to a unitary representation of the symmetry group $G = \mathbb{Z}_2$ on $\mathbb{H}_v$, the local Hilbert space of each vertex. Now suppose we are given a many-body quantum state $|\psi\rangle \in \mathbb{H}$ such that $U_\Lambda |\psi\rangle = |\psi\rangle$, i.e., one which is invariant under the global action $U_\Lambda(g) = \otimes_{v \in \Lambda} U_v(g)$ of elements $g \in \mathbb{Z}_2$. In order to promote this state into one which is invariant under the *local* action of the symmetry, we introduce new "gauge" degrees of freedom on the edges $e$ of $\Lambda$. For $G = \mathbb{Z}_2$, this corresponds to introducing gauge qubits on each edge, such that the Hilbert space on each edge $H_e = \mathbb{C}^2$, spanned by the basis $\{|0\rangle, |1\rangle\}$. Group multiplication is then given simply by the Pauli-$X$ operator: $X_e |0\rangle_e = |1\rangle_e$, $X_e |1\rangle_e = |0\rangle_e$. The Pauli-$Z$ operator acts as the nontrivial character $Z_e |0\rangle_e = |0\rangle_e$, $Z_e |1\rangle_e = -|1\rangle_e$, and $Y = iXZ$.

The Hilbert space of the combined gauge-matter system is then $\mathbb{H} := \mathbb{H}^M \otimes \mathbb{H}^G$, where the gauge Hilbert space $\mathbb{H}^G := \otimes_{e \in \Lambda} \mathbb{H}_e$. Physical states in the gauge theory must be invariant under local gauge transformations at each vertex, which correspond to the unitary operator $U_v(g) \otimes_{e \ni v} X_e$ (here $e \ni v$ denotes all edges $e$ connected to vertex $v$). We can construct a projector onto states satisfying the Gauss' law at a vertex $v$:

$$P_v := \frac{1}{2}(\mathbb{1} + U_v \prod_{e \ni v} X_e), \tag{1}$$

where $U_v := U_v(g = 1)$. Since $[P_v, P'_v] = 0 \; \forall v, v' \in \Lambda$, the projector onto the gauge-invariant subspace is defined as

$$P := \prod_v P_v. \tag{2}$$

Analogously, we must also define projectors for operators $\mathcal{O}$; for any operator $\mathcal{O}$ with nontrivial support on a compact region $\Gamma \subseteq \Lambda$, we define the operator map

$$\mathcal{P}_\Gamma[\mathcal{O}] := \frac{1}{2^{|\Gamma_0|}} \sum_{\substack{\{i_v\} \\ v \in \Gamma}} \prod_{v \in \Gamma} U_v^{i_v} \prod_{\substack{e \in \Gamma \\ v \in e}} X_e^{i_v} \; \mathcal{O} \prod_{v \in \Gamma} U_v^{i_v} \prod_{\substack{e \in \Gamma \\ v \in e}} X_e^{i_v}, \tag{3}$$

which produces gauge-invariant operators. In the above, $i_v = 0, 1$.

The prescription for gauging many-body quantum states is then given by a linear map $G : \mathbb{H}^M \to \mathbb{H}$ defined as

$$G |\psi\rangle := P |\psi\rangle \bigotimes_{e \in \Lambda} |0\rangle_e, \tag{4}$$

which embeds states $|\psi\rangle \in \mathbb{H}^M$ into the total gauge-matter Hilbert space $\mathbb{H}$. Similarly, we can define an operator gauging map $\mathcal{G}_\Gamma : \mathbb{L}(\mathbb{H}_\Gamma^M) \to \mathbb{L}(\mathbb{H}_\Gamma)$ by

$$\mathcal{G}[\mathcal{O}] := \mathcal{P}_{\Gamma(\mathcal{O})}[\mathcal{O} \bigotimes_{e \in \Gamma} |0\rangle \langle 0|_e], \tag{5}$$

which maps a matter operator $\mathcal{O}$ supported on $\Gamma \subseteq \Lambda$ onto gauge-invariant operators supported on the same region. The gauging maps satisfy the following useful identity

$$\mathcal{G}[\mathcal{O}]G = G\mathcal{O} \tag{6}$$

for any symmetric operator $\mathcal{O}$ on the matter degrees of freedom.

Gauging a matter Hamiltonian $H_M = \sum_{v,i} h_v^{(i)}$ results in a gauge and matter Hamiltonian

$$H = H_M^{(\mathcal{G})} + \Delta_{\mathcal{B}} H_{\mathcal{B}} + \Delta_P H_P \,, \tag{7}$$

where

$$H_M^{(\mathcal{G})} = \sum_{v,y} \mathcal{G}[h_v^{(i)}] \tag{8}$$

is the gauged matter Hamiltonian, the term

$$H_{\mathcal{B}} = \sum_p \left(\mathbb{1} - \mathcal{B}_p\right) \tag{9}$$

consists of zero flux constraints

$$\mathcal{B}_p = \prod_{e \in \partial p} Z_e \,, \tag{10}$$

and the term

$$H_P = \sum_v (\mathbb{1} - P_v) \tag{11}$$

penalizes gauge symmetry violations. The terms in the Hamiltonian satisfy

$$[H_M^{(\mathcal{G})}, H_{\mathcal{B}}] = [H_M^{(\mathcal{G})}, H_P] = [H_{\mathcal{B}}, H_P] = 0 \,. \tag{12}$$

In the limit $\Delta_P \to \infty$, the gauge symmetry is strictly enforced.

Due to Eq. (6), any symmetric ground state $|\psi_0\rangle$ is mapped to a ground state of the gauged Hamiltonian $G|\psi_0\rangle$. It was shown in Ref. [84] that the process of gauging a local Hamiltonian with at least one symmetric ground state preserves the existence of a spectral gap. It was also shown that the full ground space of the gauged Hamiltonian is spanned by the set of gauged symmetric states from each of the distinct symmetry-twisted sectors. For layer swap symmetry, there are 8 symmetry-twisted sectors on the 3D torus given by the choice of a layer swap domain wall on each 2-cycle. For example, with a symmetry twist on the $xy$-plane 2-cycle of an $L_x \times L_y \times L_z$ torus, the symmetric subspace is given by the subspace of the $L_x \times L_y \times 2L_z$ torus that is symmetric under swapping sites separated by $L_z$ sites in the $z$ direction. It is beyond the scope of the current paper to compute the full ground space degeneracy for the fracton models considered in this paper, but the general principles mentioned here still apply.

### 2.1.1 Disentangling

For matter transforming under the left regular representation $U_v(1) = X$ it is possible to apply a local unitary transformation to disentangle degrees of freedom that are fixed out due to the gauge constraints, thus leaving an unconstrained Hilbert space. The unitary is given by $C_\Lambda = \prod_v C_v$ where

$$C_v = \prod_{e \ni v} CX_{v,e} \,, \tag{13}$$

and $CX_{v,e}$ denotes a controlled-X operator, defined by

$$CX_{v,e}X_v CX_{v,e}^\dagger = X_v X_e, \qquad\qquad CX_{v,e}X_e CX_{v,e}^\dagger = X_e,$$
$$CX_{v,e}Z_e CX_{v,e}^\dagger = Z_v Z_e, \qquad\qquad CX_{v,e}Z_v CX_{v,e}^\dagger = Z_v. \qquad (14)$$

The disentangling map satisfies $C_\Lambda P_v C_\Lambda = \frac{1}{2}(\mathbb{1} + X_v)$. Hence, the resulting constraints fix out the $\mathbb{H}^M$ qubits and leave an unconstrained Hilbert space isomorphic to $\mathbb{H}^G$.

In fact, the gauging and disentangling steps can be carried out in a single step by following the simple recipe

$$X_v \mapsto \prod_{e \ni v} X_e, \quad Z_u Z_v \mapsto \prod_{e \in \langle u,v \rangle} Z_e, \qquad (15)$$

where $\langle u, v \rangle$ is a path between vertices $u$ and $v$. We remark that the above recipe is not unique for pairs of $Z$ operators as it requires a choice of path. However, for pairs of $Z$ operators and paths chosen within a ball, the choices only differ by $\mathcal{B}_p$ operators. Hence all choices act the same way within the ground space of $H_{\mathcal{B}}$, which consists of flat gauge connections.

## 2.2 Emergent quasi-particle description

Gauging a global $\mathbb{Z}_2$ symmetry that acts as layer swap on an emergent anyon theory $\mathcal{M} \boxtimes \mathcal{M}$, where $\mathcal{M}$ is a modular tensor category, leads to a new theory $(\mathcal{M} \boxtimes \mathcal{M})/\mathbb{Z}_2$ in the terminology of Ref. [79]. To understand the gauged theory, we first consider the symmetry-enriched theory obtained by introducing twist defects (see Fig. 1) into the $\mathcal{M} \boxtimes \mathcal{M}$ theory [82]. The $\mathbb{Z}_2$-domain walls permute the layers, and hence the symmetry defects occurring at the ends of a domain wall are labelled by a single copy of $\mathcal{M}$ corresponding to the possible eigenvalues under braiding with each symmetric anyon $aa$, which can be thought of as taking a single $a \in \mathcal{M}$ around the defect through both layers in a single closed loop. For an on-site symmetry it is simple to introduce these domain walls and defects on the lattice [84, 85]. We also note that gauging the layer exchange symmetry is equivalent to condensing (or proliferating) the domain walls, thus deconfining the twist defects. In cases where the symmetry is implemented by a translation, these topological defects correspond to familiar pointlike or looplike lattice dislocations [86].

Under the symmetry action, anyons of the form $aa$ are fixed while $ab \leftrightarrow ba$ form orbits of length two. Therefore, upon gauging the symmetry the anyons split and coalesce respectively:

$$aa \mapsto [aa, \pm], \quad \{ab, ba\} \mapsto [ab] = [ba]. \qquad (16)$$

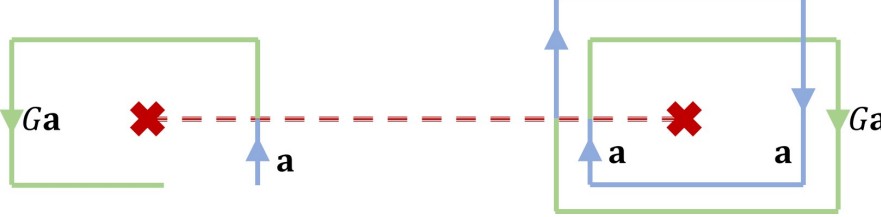

Figure 1: A $G = \mathbb{Z}_2$ twist defect switches the layer index, which can no longer be assigned globally due to the topological obstruction, but can still be assigned locally. Operators which pass through the defect cannot pass through an odd number of times and form closed loops without incurring an energy penalty. Closed loop operators stabilizing the ground space are hence required to wind around the defect line an even number of times.

| SET | Gauged model | Quant. dim. |
|---|---|---|
| Symmetric q.p. $aa$ | $[aa,+]$ | 1 |
| Permuted q.p. $1a, a1$ | $[1a]$ | 2 |
| Global $\mathbb{Z}_2$ charge | Gauge charge $[11,-]$ | 1 |
| $\mathbb{Z}_2$ Defect line | Gauge flux | Shape dependent |

Table 1: The effect of gauging a $\mathbb{Z}_2$ symmetry that acts via swapping layers in 3D on a generating set of particles. The pointlike quasi-particles (q.p.) in the first two rows may include fractons, lineons, planons, or have no mobility restrictions.

Prior to gauging, the domain walls are given by a line where layers are interchanged, so the twist defects at the ends of these lines are in one-to-one correspondence with anyons in a single layer $g_a$ and are swap symmetric. The symmetric anyons $aa$ can condense on the symmetry defects at the endpoints of the layer swap domain walls. Upon gauging, each defect splits

$$g_a \mapsto [g_a, \pm], \tag{17}$$

into a version with $\pm 1$ gauge charge. The gauged bare defect obeys non-Abelian fusion rules

$$[g_1,+] \times [g_1,+] = \sum_a [aa,+]. \tag{18}$$

We remark that one could similarly consider gauging other $\mathbb{Z}_2$ anyon permutation symmetries which swap e.g., $e$ with $m$ in a single toric code layer [86]. As a step in this direction for fracton models, Ref. [87] considered a kind of pointlike twist defect appearing at the end of a defect line in the 3D checkerboard model corresponding to its electromagnetic duality—while condensing these point defects (equivalently, gauging the symmetry) would not introduce any looplike excitations, it does offer a route to realizing subdimensional excitations (which would necessarily be lineons) with non-integer quantum dimension. Here, however, we restrict ourselves to gauging global layer swap/hadamard symmetry as described above. So as to keep our discussion self-contained, and as a review, in the appendices we discuss the effects of gauging on three familiar models: the 2D and 3D toric codes (Appendix A), and the 2D color code (Appendix B), which is equivalent via a local unitary to two copies of the 2D toric code [88–90].

### 2.3 Generating non-Abelian particles by gauging layer swap symmetry in 3D

The result of gauging layer swap symmetries upon pointlike particles in 3D follows the gauging of layer swap on anyons in 2D closely. In particular, for pointlike superselection sectors in a bilayer system, gauging is still described by Eq. (16), with non-Abelian gauged particles $[ab]$. Unlike in 2D, the behavior of the defect sector after gauging can be separated from that of the pointlike particles, since the defects all result in looplike excitations after gauging. The rest of this section contains the central ideas of this paper, as we obtain general results regarding the nature of excitations and their associated Wilson operators within the framework developed thus far. The results of this analysis are summarized in Table 1, where the possible excitations in the gauged theory, alongside their precursors in the symmetry enriched phase and their quantum dimensions, are tabulated.

Our arguments are based on fractal operators that create the pointlike excitations at their corners. Such operators have been shown to exist for all translation invariant Pauli models in 3D [91], and include the special cases of string and planar subsystem operators. For the remainder of this section we focus on gauging layer swap symmetry on theories with Abelian particles that have $\mathbb{Z}_2$ fusion rules, such as those occuring in topological stabilizer models.

The results should easily generalize to gauging symmetries that act as swap on more general Abelian theories, while the ideas are relevant for totally general quasi-particle permuting symmetry-enriched phases.

Let $\mathcal{O}_a$ denote a fractal operator that creates a cluster of well separated topologically nontrivial point excitations, which include an excitation $a$ at the origin. Since the symmetrized version of the above operator is given by $\mathcal{O}_a \otimes \mathbb{1} + \mathbb{1} \otimes \mathcal{O}_a$, the gauged version of this operator creates a pattern of excitations located identically to those created by the original operator $\mathcal{O}_a$. On the other hand, we can consider the antisymmetrization $\mathcal{O}_a \otimes \mathbb{1} - \mathbb{1} \otimes \mathcal{O}_a$. While this operator becomes 0 after gauging, as it is odd under the global symmetry, we can consider the product:

$$(\mathcal{O}_a \otimes \mathbb{1} - \mathbb{1} \otimes \mathcal{O}_a) \, T_{\vec{v}} [\mathcal{O}_a \otimes \mathbb{1} - \mathbb{1} \otimes \mathcal{O}_a],$$

where $T_{\vec{v}}$ denotes translation by a lattice vector $\vec{v}$ large compared to the support of $\mathcal{O}_a$. Gauging this operator produces a pair of operators in the gauged theory with support near that of the ungauged operators, that are additionally joined by a string of $Z$ operators on the gauge degrees of freedom. This gauged operator again creates a charge pattern at the same location as that created by the ungauged theory. Intuitively, this argument shows that each of the charge cluster operators carries a gauge charge.

To make the above more precise, let us now consider how one locally measures gauge charge, as well as the related concept of locally measuring $\mathbb{Z}_2$ symmetry charges before gauging. Gauge charge in a region $\mathcal{R}$ containing the origin is measured by braiding a looplike gauge flux excitation over $\partial \mathcal{R}$, with the operator implementing this process given by gauging a restriction of the global symmetry transformation applied to $\mathcal{R}$. Such a restriction is obtained by applying the global symmetry to $\mathcal{R}$, followed by the application of an operator $\overline{\mathcal{V}}_{\partial \mathcal{R}}$ that removes the domain wall thus created at the boundary by braiding a defect line over $\partial \mathcal{R}$,

$$U_{\mathcal{R}} = \overline{\mathcal{V}}_{\partial \mathcal{R}} \prod_{v \in \mathcal{R}} U_v, \tag{19}$$

with this process illustrated in Fig. 2. Gauging the above operator results in an operator supported only near $\partial \mathcal{R}$. We remark that the restriction of the symmetry satisfies $U_{\mathcal{R}} |\psi_0\rangle = d |\psi_0\rangle$ on the symmetric ground state, for some positive $d$, by construction.

If we now consider a nonzero ground state of the gauged system, given by $G |\psi_0\rangle$, we may create an excitation and braid a flux loop around it as follows:

$$\mathcal{G}[U_{\mathcal{R}}] \mathcal{G}[\mathcal{O}_a \otimes \mathbb{1} + \mathbb{1} \otimes \mathcal{O}_a] G |\psi_0\rangle = G U_{\mathcal{R}} (\mathcal{O}_a \otimes \mathbb{1} + \mathbb{1} \otimes \mathcal{O}_a) |\psi_0\rangle, \tag{20}$$

which follows from Eq. (6). Assuming that $\mathcal{O}_a$ is a tensor product of local operators, which holds for stabilizer models, we then have that

$$\begin{aligned} U_{\mathcal{R}} \, \mathbb{1} \otimes \mathcal{O}_a |\psi_0\rangle &= \overline{\mathcal{V}}_{\partial \mathcal{R}} \, \mathcal{O}_a^{(\mathcal{R})} \otimes \mathcal{O}_a^{(\mathcal{R}^c)} \prod_{v \in \mathcal{R}} U_v |\psi_0\rangle \\ &= \overline{\mathcal{V}}_{\partial \mathcal{R}} \, \mathcal{O}_a^{(\mathcal{R})} \otimes \mathcal{O}_a^{(\mathcal{R}^c)} \mathcal{V}_{\partial \mathcal{R}} |\psi_0\rangle, \end{aligned} \tag{21}$$

where $\mathcal{V}_{\partial \mathcal{R}}$ creates a domain wall by braiding a defect over $\partial R$. However, this process *must leave an excitation* on $\partial R$, since $\mathcal{O}_a^{(\mathcal{R})} \otimes \mathcal{O}_a^{(\mathcal{R}^c)}$ creates a pair of excitations over both layers, in the same superselection sector as $aa$ and with support on $\partial R \cap \text{supp} \, \mathcal{O}_a$; this is depicted schematically in Fig. 2 (b). This configuration cannot be deleted by the combined action of $\overline{\mathcal{V}}_{\partial \mathcal{R}}, \mathcal{V}_{\partial \mathcal{R}}$. Therefore, this braiding process in the gauged theory *cannot* result in fusion of the flux loop to the vacuum or, in other words, the matrix element of the braiding is zero. Hence, there is no well defined $\mathbb{Z}_2$ charge on the gauged $[ab]$ excitation when $a \neq b$. We remark that for a gauged $aa$ particle there is no such obstruction, and so assigning a $\mathbb{Z}_2$ charge to the gauged particle $[aa, \pm]$ is well-defined.

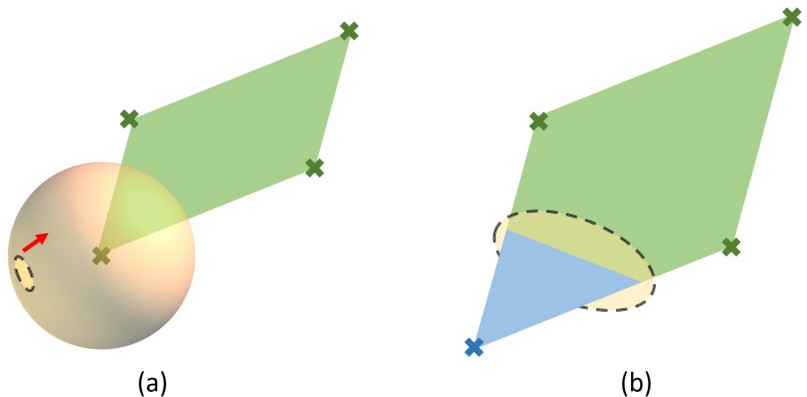

(a)              (b)

Figure 2: The process of (a) applying the symmetry locally to permute a charge and then removing the domain wall thus created by dragging a twist loop over it cannot result in the twist loop fusing to vacuum, since it results in the configuration shown in (b). Operators acting on the first (second) layer are represented by the green (blue) region.

Combined with the above observation regarding gauging a pair of charge clusters created by $(\mathcal{O}_a \otimes \mathbb{1} - \mathbb{1} \otimes \mathcal{O}_a) T_{\vec{v}}[\mathcal{O}_a \otimes \mathbb{1} - \mathbb{1} \otimes \mathcal{O}_a]$, we see that each cluster of charges can fuse to either $\pm$ gauge charge, which cannot be detected *locally* on a single excitation $a$. For a line of $N$ such charge clusters with a vacuum total charge, there are thus $2^{N-1}$ possible internal charge configurations.

In fact, we can show that the quantum dimension of an individual gauged excitation $[1a]$ is 2, provided that $a$ obeys $\mathbb{Z}_2$ fusion with itself. Since the $\mathbb{Z}_2$ charge of individual $[1a]$ particles is not well defined, the operator

$$\mathcal{G}[(\mathcal{O}_a \otimes \mathbb{1} + \mathbb{1} \otimes \mathcal{O}_a) \prod_{i=0}^{N}(\mathbb{1} + c_i)],\tag{22}$$

should create the same pattern of local excitations in the vicinity of $\mathcal{O}_a$ as that created by $\mathcal{G}[\mathcal{O}_a \otimes \mathbb{1} + \mathbb{1} \otimes \mathcal{O}_a]$. Here, $i = 1, \dots, N$ label the locations of the $N$ well separated charges created by $\mathcal{O}_a$, $i = 0$ labels some other point far from the support of $\mathcal{O}_a$, and $c_i$ are some local operators with negative charge under the global $\mathbb{Z}_2$ symmetry.

To finish this argument, notice that since

$$(\mathcal{O}_a \otimes \mathbb{1} + \mathbb{1} \otimes \mathcal{O}_a)^2 = 2(\mathbb{1} + \mathcal{O}_a \otimes \mathcal{O}_a),\tag{23}$$

fusing the same local pattern of $[1a]$ charges created in the two distinct ways described above results in a superposition of vacuum and $N$ $[aa, +]$ charges, together with $2^N$ configurations of gauge charges. Since all the aforementioned particles are Abelian, we have that

$$2d_{[1a]}^N = 2^{N+1},\tag{24}$$

i.e. $d_{[1a]} = 2$. Technically, this argument required that the superselection sectors of each of the $N$ particles created by $\mathcal{O}_a$ are related by translation (or are isomorphic in a more general sense) which holds for all examples considered below. This completes our proof that pointlike particles obeying $\mathbb{Z}_2$ fusion in the symmetry enriched phase lead to non-Abelian particles $[1a]$ in the gauged phase. We remark that nowhere in the preceding argument have we made any assumptions about the *mobility* of the $a$ particles. Thus, we have shown that gauging a symmetry which permutes subdimensional excitations will result in non-Abelian excitations in the gauged phase.

However, to show that gauging a fracton permuting symmetry results in non-Abelian *sub-dimensional* excitations, we need to show that the gauged phase remains fractonic *i.e.,* that the gauging procedure does not produce any stringlike operators at the ends of which fractons may be isolated. We now show that subdimensional excitations indeed retain their mobility restrictions upon gauging.

A gauge invariant operator that commutes with the flux constraints and is supported within a ball can be written as

$$O = \sum_i \mathcal{O}_i^M \otimes \mathcal{O}_i^G \,, \tag{25}$$

where

$$\mathcal{O}_i^G = \prod_j \prod_{e \in \partial B_j^i} X_e \,, \tag{26}$$

is the product of $X$ matrices on edges through the boundary on the dual lattice of some collection of balls $B_j^i$. On the gauged subspace this satisfies

$$OG = P_\Lambda O \bigotimes_e |0\rangle_e$$

$$= G \sum_i \prod_j \left( \prod_{v \in B_j^i} U_v \right) \mathcal{O}_i^M \,. \tag{27}$$

Assuming that $O$ only created a pair of pointlike excitations on the groundstate $G|\psi\rangle$, then away from those excitations $\mathcal{O}_i^M$ must contain domain wall operators along $\partial B_j^i$ so as to avoid a growing energy penalty. These $\mathbb{Z}_2$ domain walls can then be dragged into a small neighborhood of the excitations, thereby removing the $U_v$ terms they are dragged over. This process results in a new operator $\widetilde{\mathcal{O}}^M$ that creates a pair of pointlike excitations on the ungauged layers, which map onto the original pair of excitations after gauging. Since the operator gauging map is invertible on the subspace of symmetric operators, this implies a pair creation operator for any particles that are mapped to the aforementioned pair after gauging. In other words, fractons are mapped to fractons after gauging—more precisely, fractons created at the corners of fractal (planar) operators map onto fractons created at the corners of fractal (planar) operators after gauging, as the operator gauging map preserves support.

Combined with our discussion showing that pointlike excitations of the form $[1a]$ have quantum dimension $d_{[1a]} = 2$, the above argument clearly establishes the presence of non-Abelian fractons in the phase obtained by gauging a fracton permuting symmetry. The same argument applies to other subdimensional excitations permuted by the symmetry as well. Furthermore, these fractons are *inextricably* non-Abelian, *i.e.,* they are not the composite of an Abelian fracton with a non-Abelian particle that is not a fracton, as long as the ungauged particle $1a$ is a true non-composite fracton. This is due to the fact that the only Abelian fractons in the gauged theory are of the form $[bb, \pm]$; for $b = a$, this fracton can be absorbed by the $[1a]$ fracton, since otherwise it leads to a distinct non-Abelian fracton $[b(ab)]$. For theories also containing subdimensional excitations that are not fractons, so long as a fracton $1a$ in the ungauged theory is not a composite, the gauged particle is also not a composite. Since all models we consider host non-composite fractons, the above argument ensures the presence of inextricably non-Abelian fractons in the gauged theory.

Finally, to exhaust the set of deconfined excitations in the gauged phase, we discuss the gauge fluxes. Similarly to the 2D case discussed in Sec. 2.2, the line defects occurring at the boundary of layer swap surfaces can absorb symmetric point particles $aa$ incident on them.

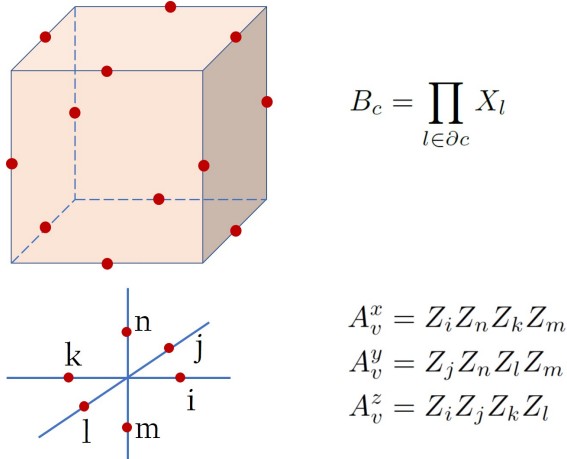

$$B_c = \prod_{l \in \partial c} X_l$$

$$A_v^x = Z_i Z_n Z_k Z_m$$
$$A_v^y = Z_j Z_n Z_l Z_m$$
$$A_v^z = Z_i Z_j Z_k Z_l$$

Figure 3: The X-Cube model [8] is defined on a cubic lattice with a qubit on each link. The Hamiltonian is a sum of commuting terms, with $B_c$ and $A_v^k$ the cube and vertex stabilizers respectively.

For fracton orders, the existence of a subextensive number of superselection sectors leads to a large and shape dependent quantum dimension depending upon the number of independent topological sectors of $aa$ particles supported along the loop. A similar effect persists after gauging, with the addition of a possible gauge charge on the loop excitation and each $[aa, \pm]$ particle. This completes our general discussion of gauging $\mathbb{Z}_2$ permutation symmetries in 3D topological or fractonic lattice models, with the results summarized in Table 1.

## 3 Examples

### 3.1 Two Copies of the X-Cube Model with Layer Exchange Symmetry

Here, we consider the X-Cube model introduced in Ref. [8], starting with a brief review of its properties so as to keep our discussion self-contained. The X-Cube model is defined on a cubic lattice with a single qubit living on each link. The Hamiltlonian is a sum of commuting terms

$$H_{XC} = -\sum_{v,k} A_v^k - \sum_c B_c, \tag{28}$$

where $k = x, y, z$. The first term $A_v^k$ is a product of four Pauli-$Z$ operators acting on the links in the plane perpendicular to direction $k$, while the second term is a product over twelve Pauli-$X$ operators acting on the links forming an elementary cube of the lattice (see Fig. 3). The Hamiltonian $H_{XC}$ is exactly solvable, with any ground state $|\Phi\rangle$ satisfying

$$A_v^k |\Phi\rangle = |\Phi\rangle, \quad B_c |\Phi\rangle = |\Phi\rangle, \quad \forall v, k, c. \tag{29}$$

On an $L_x \times L_y \times L_z$ three-torus, the models has $2^{2(L_x+L_y+L_z)-3}$ locally indistinguishable ground states, with the sub-extensive ground state degeneracy a characteristic feature of 3D gapped fracton models.

Excitations are created by flipping the eigenvalue of at least one of the terms in the Hamiltonian, with no local operator present which creates only a single excitation. Fractons, in particular, are created from the ground state by flipping the eigenvalue of a cubic interaction term from $+1$ to $-1$; unlike anyons in ordinary topological orders, created at the ends of

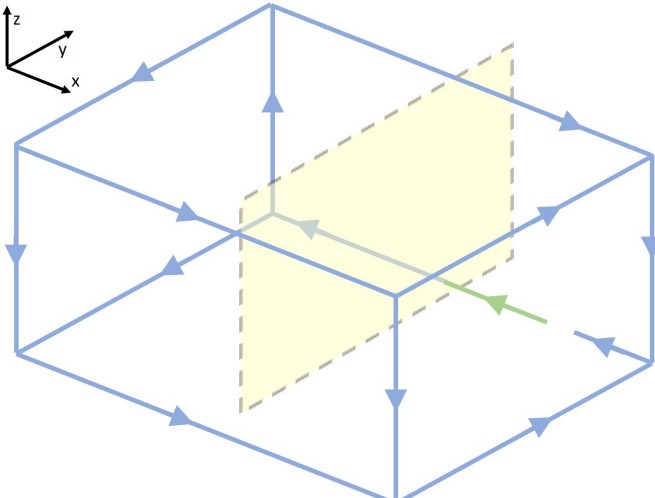

Figure 4: A layer swap domain wall (in yellow) swaps a lineon from the first X-cube layer (green) to the second layer (blue). Clearly operators which pass through the domain wall an odd number of times cannot form closed cages and so incur an energy penalty. Cage operators that stabilize the ground space are hence required to wind around the defect surface an even number of times.

Wilson strings, fractons are created at the corners of Wilson membranes, leading to the immobility of isolated fractons. Nevertheless, bound states of adjacent fractons are planons as they can be separated at the ends of Wilson strings and are hence mobile along planes of the 3D system. Excitations can also be created by changing the eigenvalue of the $A_v^k$ terms in the Hamiltonian—these are mobile only along one-dimensional lines of the lattice.

Given that each fracton constitutes its own superselections sector [8], a particularly convenient way of characterizing excitations in the X-Cube model is through its *quotient* superselection sectors (QSS)[5]. Specifically, a QSS is defined as the class of topologically non-trivial excitations that are equivalent to each other through local unitary operations and attaching planons. In other words, the QSS is effectively the number of superselection sectors modulo planons.

Considering two copies of the X-Cube model, we can label the generators of the QSS by the ordered pair $ab$, where $a(b) = f, l_x, l_y$ labels fractons, lineons mobile along $x$, and lineons mobile along $y$ in the first (second) X-Cube copy. Other QSS can be generated through e.g., the fusion rule $l_x 1 \times l_y 1 = l_z 1$ (and similarly for the other layer), such that the six elementary QSS generators give rise to total $2^6 = 64$ QSS.

Before we gauge the layer exchange symmetry, let us consider the effect of introducing a symmetry defect into the system. As discussed in the previous section, in 3D we can introduce layer swap surfaces which provide a topological obstruction to the global definition of layer index. To see this, imagine passing an $l_x 1$ lineon through the layer swap domain wall, as shown in Fig. 4, which causes the layer index of the lineon to change. Unlike fully mobile excitations, the lineon cannot simply loop around the symmetry defect to return to its starting position; instead, the $1l_x$ loop splits into $1l_y$ and $1l_z$ lineons and returns to its initial position by forming the depicted cage configuration. As should be evident, for the cage to close, the

---

[5]The notion of quotient super-selection sectors was introduced in Ref. [92] to characterize excitations in "foliated" fracton phases, which provide a coarse-grained notion of equivalence classes of fracton orders: Two fracton phases A and B are considered equivalent as foliated fracton phases if A stacked with decoupled layers of 2D topologically ordered states is adiabatically connected to B, stacked with a possibly different set of 2D topologically ordered layers. It remains unclear whether this notion extends to non-Abelian fracton orders [62, 75, 76].

lineon needs to pass through the domain wall an even number of times so as to not incur an energy penalty.

Similar arguments apply to the other QSS generators as well, illustrating that the domain wall can absorb the generating particles of the form $cc$. However, recall that the QSS are defined only modulo planons and that, strictly speaking, the fractons and lineons constitute sub-extensively many superselection sectors labelled by their positions. Therefore, unlike the usual case of defect loops in 3D topological orders, where the loop can absorb a finite number of topological excitations [93], the looplike defects in fracton orders are qualitatively distinct. Specifically, we expect that the topological degeneracy associated with genon loops in fracton orders will grow with the number of linearly independent $cc$ sectors supported along the loop. While the precise degeneracy will depend on the specific fracton model being considered, its extensive nature is a generic feature of fracton orders with an on-site $\mathbb{Z}_2$ global symmetry.

Now, before we gauge the layer exchange symmetry between the two copies of the X-Cube model, we regroup the qubits from the edges of the cubic lattice to the vertices, as shown in Fig. 5. After regrouping, there are six qubits per site, with the Hamiltonian given by

$$
\begin{aligned}
H_{2\times XC} = &-\sum_c \mathcal{X}_c - \sum_c \mathcal{Z}_c^x - \sum_c \mathcal{Z}_c^y - \sum_c \mathcal{Z}_c^z \\
&-\sum_c \widetilde{\mathcal{X}}_c - \sum_c \widetilde{\mathcal{Z}}_c^x - \sum_c \widetilde{\mathcal{Z}}_c^y - \sum_c \widetilde{\mathcal{Z}}_c^z,
\end{aligned}
\tag{30}
$$

with the usual cube and vertex terms of the X-Cube model[6] now given by:

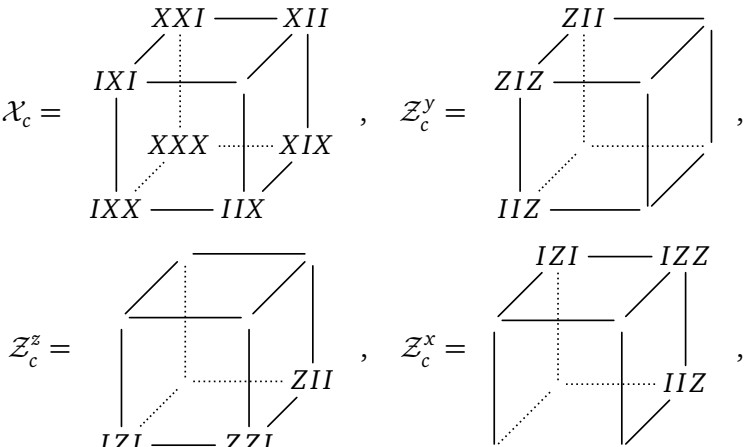

which all act on the first X-Cube layer. The terms acting on the second layer are obtained by taking $\left(X \to \widetilde{X}, Z \to \widetilde{Z}\right)$ in the above.

---

[6]We are mapping the terms of the original X-Cube model, defined in Fig. 3 as $B_c \to \mathcal{X}_c$ and $A_v^k \to \mathcal{Z}_c^k$ for $k = x, y, z$

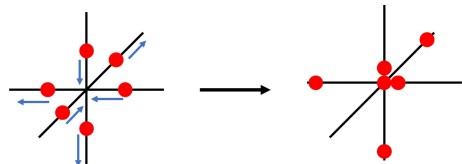

Figure 5: Qubit degrees of freedom originally living on the edges of the cubic lattice are regrouped onto the vertices, such that three qubits are grouped onto each vertex for a single X-Cube layer.

The on-site $\mathbb{Z}_2$ symmetry under which the above Hamiltonian is invariant is generated by

$$U_\nu = \text{SWAP} \otimes \text{SWAP} \otimes \text{SWAP}, \tag{31}$$

which acts as follows on each site $r = (x, y, z)$ of the cubic lattice:

$$X^{\hat{i}}_{x,y,z} \leftrightarrow \widetilde{X}^{\hat{i}}_{x,y,z}, \quad Z^{\hat{i}}_{x,y,z} \leftrightarrow \widetilde{Z}^{\hat{i}}_{x,y,z}, \tag{32}$$

where superscritpt $\hat{i} = \hat{x}, \hat{y}, \hat{z}$ labels the edge from whence the qubit was displaced on to the vertex $\nu$ at position $x, y, z$. We can now follow the general procedure for gauging global on-site $\mathbb{Z}_2$ symmetries, as described in Sec. 2. We introduce an additional gauge qubit on each link $(x + 1/2, y, z), (x, y + 1/2, z), (x, y, z + 1/2)$ of the cubic lattice and gauge the Hamiltonian to obtain

$$H = -\sum_c \mathcal{G}\left[\mathcal{X}_c + \widetilde{\mathcal{X}}_c\right] - \sum_c \mathcal{G}\left[\mathcal{Z}^x_c + \widetilde{\mathcal{Z}}^x_c\right] - \sum_c \mathcal{G}\left[\mathcal{Z}^y_c + \widetilde{\mathcal{Z}}^y_c\right] - \sum_c \mathcal{G}\left[\mathcal{Z}^z_c + \widetilde{\mathcal{Z}}^z_c\right]$$
$$+ H_{\mathcal{B}} + \Delta_P H_P, \tag{33}$$

where the first term acts on gauge qubits on all twelve links forming a cube, which we list for completeness: $(x+1/2, y, z), (x, y+1/2, z), (x+1/2, y+1, z), (x+1, y+1/2, z), (x, y, z+1/2),$ $(x+1, y, z+1/2), (x, y+1, z+1/2), (x+1, y+1, z+1/2), (x+1/2, y, z+1), (x, y+1/2, z+1),$ $(x+1, y+1/2, z), (x+1/2, y+1, z+1)$. Meanwhile, the second term acts on qubits at edges $(x, y-1/2, z), (x, y, z-1/2)$, the third on $(x-1/2, y, z), (x, y, z-1/2)$, and the fourth on $(x-1/2, y, z), (x, y-1/2, z)$.

The term $H_{\mathcal{B}}$ is given by

$$H_{\mathcal{B}} = -\sum_{x,y,z} Z_{x+\frac{1}{2},y,z} Z_{x,y+\frac{1}{2},z} Z_{x+\frac{1}{2},y+1,z} Z_{x+1,y+\frac{1}{2},z}$$
$$-\sum_{x,y,z} Z_{x,y+\frac{1}{2},z} Z_{x,y,z+\frac{1}{2}} Z_{x,y+\frac{1}{2},z+1} Z_{x,y+1,z+\frac{1}{2}}$$
$$-\sum_{x,y,z} Z_{x+\frac{1}{2},y,z} Z_{x,y,z+\frac{1}{2}} Z_{x+\frac{1}{2},y,z+1} Z_{x+1,y,z+\frac{1}{2}}, \tag{34}$$

and the term $H_P$ is given by

$$H_P = -\sum_{x,y,z} U_{x,y,z} X_{x+\frac{1}{2},y,z} X_{x,y+\frac{1}{2},z} X_{x,y,z+\frac{1}{2}} X_{x-\frac{1}{2},y,z} X_{x,y-\frac{1}{2},z} X_{x,y,z-\frac{1}{2}}, \tag{35}$$

up to an overall shift in energy.

With the gauging map as specified in Sec. 2, the above Hamiltonian in principle constitutes an *explicit* exactly solvable Hamiltlonian whose properties, such as the ground state degeneracy and statistical properties of excitations, can be analytically derived. While appropriate for deriving quantitative features such as the ground state degeneracy, this algebraically tedious procedure sheds no further light on the qualitative features of the gauged model than those derived from the general arguments presented in Secs. 2.2 and 2.3.

To begin with, let us consider the effect of the layer exchange symmetry on the QSS of the two copies. The SWAP symmetry acts as follows on the generators of the QSS:

$$f1 \leftrightarrow 1f, \quad l_x 1 \leftrightarrow 1l_x, \quad l_y 1 \leftrightarrow 1l_y. \tag{36}$$

Thus, particles of the form $cc$ are fixed by the symmetry while the $1c, c1$ particles form length-2 orbits, for $c = f, l_x, l_y$ (the same will also be true for the bare superselection sectors). The symmetry defects are the looplike genons appearing at boundaries of layer swap domain walls,

such as the one depicted in Fig. 4. These defects can carry a charge under each string or membrane operator associated with the symmetric subdimensional excitations.

Our discussion of the genon loops so far parallels our discussion regarding the 3D toric code, discussed in Appendix A.2. However, there is a crucial qualitative distinction between these two cases, stemming from the presence of an *extensive* number of superselection sectors in fracton models. In particular, each lineon $l_x1$ carries a position index labelling the line along which it is mobile; due to this, the number of lineons of the form $l_x l_x$ which can be absorbed by a genon loop is determined by the *geometry* of the loop—this is in stark contrast to genon loops in the 3D toric code [93] and provides yet another indication of the geometric nature of fracton order. In general, the genons will be labelled by their eigenvalues under braiding with excitations of the form $p_i p_i$, where we have restored the appropriate position indices $i$ associated with the fracton, lineon, and planon particles $p$. Distinct genons will then be related by fusion with the Abelian particles of the form $p_i1$ and can absorb $p_i p_i$. While a quantitative analysis of the degeneracy associated with genon loops is beyond the scope of this work, the above arguments suffice to demonstrate the non-Abelian, as well as geometric, nature of the symmetry defects in fracton phases.

Once we gauge the layer exchange symmetry, the vacuum, along with the QSS generators of the form $cc$ split into disctint Abelian excitations labelled by $[cc, \pm]$ for $c = f, l_x, l_y$. Simultaneously, generators of the form $c1$ and $1c$ coalesce into single non-Abelian particles $[c1]$ with quantum dimension $d = 2$. Given the presence of *Abelian* fractons in the gauged theory, in order to show that the non-Abelian fractons are *inextricably* non-Abelian [62,76], we need to exclude the possibility that the $[f1]$ excitations are bound states of Abelian fractons with some non-Abelian planons. If this were the case, then fusing an $[ff, +]$ excitation with $[f1]$ should produce a non-Abelian planon. However, since in the gauged model $[f1]$ can absorb the Abelian fractons, this possibility is precluded. At the level of operators, we can restate the above as the statement that the membrane operator $\mathcal{O}_{[f1]}$ creating non-Abelian fractons at its corners *cannot* be local unitary equivalent to an operator creating Abelian fractons at identical locations as $\mathcal{O}_{[f1]}$ times some stringlike operator. Similarly, one can show that the non-Abelian lineons in the gauged phase are not bound states of some Abelian lineon and some non-Abelian planon. This clarifies that the non-Abelian character of fractons and lineons is a fundamentally 3D feature, since the possibility that their topological degeneracy stems from non-Abelian planons has been excluded.

Gauging also introduces $\mathbb{Z}_2$ gauge charges $[11, \pm]$, which are *fully mobile* Abelian particles. The presence of these 3D pointlike particles already indicates the qualitatively distinct nature of the gauged phase from the underlying fracton order, which lacks any fully mobile particles. Finally, upon gauging, the genon loops lead to gauge flux loops, which remain non-Abelian since they can (at the very least) absorb $[cc, +]$ and $[cc, -]$, for $c = f, l_x, l_y$. However, the gauge flux loops can also absorb Abelian planons $[p_i p_i, \pm]$, which although trivial at the level of QSS, will contribute to the quantum dimension of the gauge fluxes. Thus, the number of excitations these loops can absorb will again depend on their *geometry*, such that these loops will induce a ground state degeneracy which depends on their shape. While an exact analysis of this degeneracy is beyond the scope of this paper, we stress the qualitative distinction between the loop excitations generated by gauging in the X-Cube and in the toric code model.

Also important is the fact that the loop excitations in the gauged model braid non-trivially with the subdimensional excitations, which demonstrates that the gauged Hamiltonian *cannot* be equivalent via finite-depth-local-unitaries to a some non-Abelian fracton order decoupled from some non-Abelian TQFT. Thus, the general gauging procedure described in Sec. 2, when applied to two copies of an Abelian fracton order, naturally leads to an entirely novel quantum order, distinct from both non-Abelian TQFTs and non-Abelian fracton orders[7]. Since the model

---

[7]While this is a question of semantics, fracton order as currently defined refers to systems with *only* subdimen-

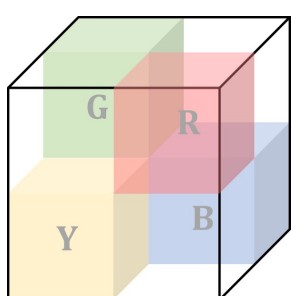

Figure 6: A $2 \times 2 \times 2$ cell of the checkerboard cubic lattice. Shaded (empty) cubes belong to the $\mathcal{A}\,(\mathcal{B})$ sublattice of the checkerboard bipartition. The $\mathcal{A}$ is further partitioned by a four-coloring into {r,g,b,y} sublattices.

we have found encompasses aspects of both fracton order (subdimensional excitations) and 3D topological order (looplike excitations), we introduce the term *panoptic fracton order* for describing such hybrid gapped 3D quantum phases.

## 3.2  Checkerboard model with on-site Hadamard symmetry

The checkerboard model, introduced in Ref. [8], is a foliated type-I fracton model defined on the 3D cubic lattice, with one qubit living on each vertex. The Hamiltonian is given by

$$H_{CB} = -\sum_{c \in \mathcal{A}} \mathcal{X}_c - \sum_{c \in \mathcal{A}} \mathcal{Z}_c, \tag{37}$$

where we have bipartitioned the cubic lattice into $\mathcal{A}$ and $\mathcal{B}$ checkerboard sub-lattices and where the sum in both terms of the Hamiltonian indexes cubes in the $\mathcal{A}$ sub-lattice. The term $X_c$ ($Z_c$) is given by the product of eight Pauli-$X$ (Pauli-$Z$) operators acting on the vertices of the cube $c$:

$$\mathcal{X}_c = \begin{matrix} X & \!\!\!-\!\!\!- & X \\ X & \!\!\!-\!\!\!- & X \\ & X & \!\!\!\cdots\!\!\! & X \\ X & \!\!\!-\!\!\!- & X \end{matrix} \quad , \quad \mathcal{Z}_c = \begin{matrix} Z & \!\!\!-\!\!\!- & Z \\ Z & \!\!\!-\!\!\!- & Z \\ & Z & \!\!\!\cdots\!\!\! & Z \\ Z & \!\!\!-\!\!\!- & Z \end{matrix} \quad .$$

This model belongs to the class of CSS-type stabilizer code Hamiltonians and is exactly solvable since it is a sum of commuting projectors, each of which is a product of Pauli operators. As can be explicitly checked, on an $L \times L \times L$ three-torus the ground state degeneracy is $2^{6L-6}$. Note that this is precisely the ground state degeneracy of two copies of the X-Cube model on a three-torus of length $L/2$—we will return to this point in Sec. 3.2.1.

In analogy with our treatment of the 2D color code (see Appendix B), to describe excitations of the checkerboard model we further partition the $a$ sublattice by introducing a four-coloring {r,g,b,y}, as shown in Fig. 6. As discussed in Ref. [94], excitations of this model can also be characterized through their QSS, as was the case with the X-Cube model. A convenient choice for the generating set for the QSS sectors is given by fracton excitations labelled by the ordered pair $c_X c_Z$, indicating the violated $\mathcal{X}_c$ ($\mathcal{Z}_c$) stabilizer in the $c_X(c_Z) = r, g, b$ sublattice. For example. $r1$ denotes a fracton excitation of a red $\mathcal{X}_c$ stabilizer. There are thus six elementary generators for the QSS, leading to a total $2^6 = 64$ QSS sectors. As discussed in Ref. [94], a pair of neighboring fractons belonging to the same sublattice constitute a planon while a

---

sional excitations and no fully mobile 3D pointlike or looplike excitations.

pair of neighboring fractons belonging to distinct sublattices constitute a lineon, as depicted in Figs. 7(b) and (c).

The elementary QSS generators are fractons, which are created at the ends of membrane operators given by

$$X_{\mathcal{M}}^{(c)} = \prod_{v \in \mathcal{M}} X_v, \quad Z_{\mathcal{M}}^{(c)} = \prod_{v \in \mathcal{M}} Z_v, \tag{38}$$

where $\mathcal{M}$ is a flat, rectangular region connecting four cubes of the same color $c = r, g, b$; Fig. 7(a) depicts the operator $X_{\mathcal{M}}^{(r)}$ which creates four $1r$ excitations. String operators for the lineons can also be defined analogously:

$$X_{l_x}^{(r,b)} = \prod_{v \in l_x} X_v, \quad Z_{l_x}^{(r,b)} = \prod_{v \in l_x} Z_v,$$

$$X_{l_y}^{(r,y)} = \prod_{v \in l_y} X_v, \quad Z_{l_y}^{(r,y)} = \prod_{v \in l_y} Z_v,$$

$$X_{l_z}^{(r,g)} = \prod_{v \in l_z} X_v, \quad Z_{l_z}^{(r,g)} = \prod_{v \in l_z} Z_v, \tag{39}$$

where $l_j$ denotes a straight line along the $j$ axis connecting pairs of adjacent fractons with distinct colors $c_1, c_2$. The operator $X_{l_x}^{(r,b)}$, creating a $1r \times 1b$ lineon is shown in Fig. 7(b) as an example.

We now proceed to gauge the on-site $\mathbb{Z}_2$ symmetry of the checkerboard model Eq. (37), which is generated by $h^{\otimes N}$, with $N$ the number of sites of the lattice and $h$ the Hadamard matrix

$$h = \begin{pmatrix} 1 & 1 \\ 1 & -1 \end{pmatrix}. \tag{40}$$

This symmetry acts on the QSS sectors as follows

$$ab \leftrightarrow ba, \tag{41}$$

mapping e.g., $1r \leftrightarrow 1r$. As we will discuss shortly, this symmetry action is equivalent to layer exchange on the QSS of two copies of the X-Cube model, but not on the bare superselection sectors themselves. Proceeding as in the case of the 2D color code (Appendix B), we find it

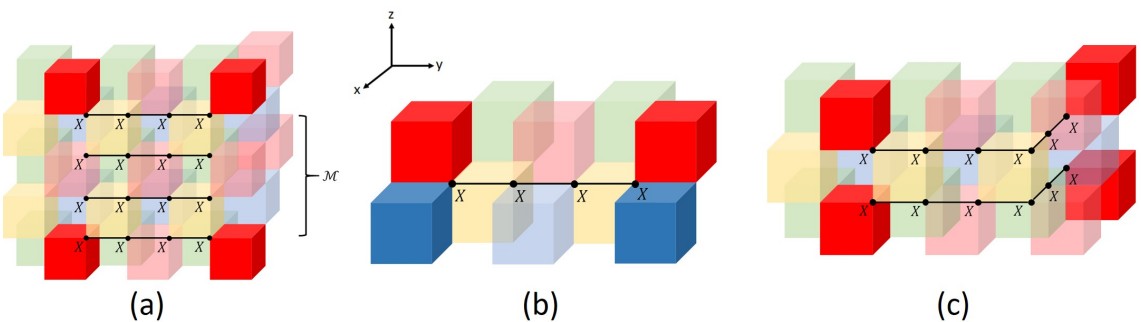

(a)          (b)          (c)

Figure 7: Hierarchy of excitations in the checkerboard model. (a) Fractons in a given sublattice are created at the corners of a membrane operator $\mathcal{M}$ as depicted. (b) Adjacent fractons belonging to distinct sublattices are lineons. (c) A pair of neighboring fractons belonging to the same sublattice is mobile along a plane and is hence a planon.

convenient to change basis: $h \mapsto X$ such that the checkerboard Hamiltonian, up to an irrelevant normalization factor, becomes

$$H = -\frac{1}{2}\sum_{c\in\mathcal{A}}\prod_{v\in\partial c}(X_v + Z_v) - \frac{1}{2}\sum_{c\in\mathcal{A}}\prod_{v\in\partial c}(X_v - Z_v) = -\sum_{c\in\mathcal{A}}\sum_{i=0}^{4}B_c^{(2i)}, \tag{42}$$

where

$$B_c^{(j)} := \sum_{s\in\mathbb{Z}_2^8,\,\text{wt}(s)=j}\prod_{v=1}^{8}X_v^{1+s_v}Z_v^{s_v}. \tag{43}$$

Here, $v = 1,\ldots 8$, label the vertices of a cube $c$ in the $\mathcal{A}$ sublattice, and $\text{wt}(s)$ is the weight function equal to the number of nonzero entries in the vector $s$. Schematically, we can expand the above formal expression out as follows:

$$H = -\,\text{[cube diagram with }X\text{]} - \text{[cube diagram with }Z\text{]} - \left(\text{[cube diagram]} + \text{permutations}\right)$$

$$-\left(\text{[cube diagram]} + \text{permutations}\right) - \left(\text{[cube diagram]} + \text{permutations}\right). \tag{44}$$

The symmetrized membrane operators creating the QSS generating fractons $cc$ and $c1+1c$ are given by

$$Y_{\mathcal{M}}^{(c)} = \prod_{v\in\mathcal{M}}Y_v,$$
$$S_{\mathcal{M}}^{(c)} = \frac{1}{2}\prod_{v\in\mathcal{M}}(X_v + Z_v) + \frac{1}{2}\prod_{v\in\mathcal{M}}(X_v - Z_v) \tag{45}$$

respectively, for $c = r, g, b$. The operator $S_{\mathcal{M}}^{(c)}$ is given by a sum over all products of either $X_v$ or $Z_v$ on each vertex along the membrane $\mathcal{M}$, with the constraint that the number of $Z_v$ operators in the membrane must be even. Similarly, we can defined symmetrized string operators for the lineons, e.g.,

$$Y_{l_x}^{(r,b)} = \prod_{v\in l_x}Y_v,$$
$$S_{l_x}^{(r,b)} = \frac{1}{2}\prod_{v\in l_x}(X_v + Z_v) + \frac{1}{2}\prod_{v\in M}(X_v - Z_v), \tag{46}$$

which creates $rr \times bb$ and $1r \times 1b + r1 \times b1$ lineons respectively. Again, the operator $S_{l_x}^{(r,b)}$ is given by a sum over all products of either $X_v$ or $Z_v$ on each vertex along the line $l_x$ subject to the constraint that only an even number of $Z_v$ operators are allowed. The $(r, g)$ and $(r, y)$ symmetrized operators follow similarly.

Since we have chosen a basis where the symmetry is acting via the left regular representation we can gauge the symmetry and disentangle the gauge constraints following Eq. (15). This results in the Hamiltonian

$$H = -\sum_{c\in\mathcal{A}}\sum_{i=0}^{4}\widetilde{B}_c^{(2i)} + H_{\mathcal{B}},\tag{47}$$

for $H_{\mathcal{B}}$ given in Eq. (34) and

$$\widetilde{B}_p^{(j)} := \sum_{s\in\mathbb{Z}_2^8,\,\mathrm{wt}(s)=j} F_p^{(s)}\prod_{v=1}^{8} A_v^{1+s_v}\tag{48}$$

where $v = 1,\ldots,8$, label the vertices of the cube along a loop $\gamma$, within the edges of the cube, that passes through each vertex once, and

$$A_v = \prod_{e\ni v} X_e,\tag{49}$$

$$F_p^{(s)} = \prod_{e=1}^{8} Z_e^{\tilde{s}_e},\tag{50}$$

for $e = 1,\ldots,8$, the edges of $\gamma$ and

$$\tilde{s}_e = \sum_{i=1}^{e} s_i \mod 2.\tag{51}$$

Eq. (15) can also be applied to gauge the string and membrane operators for symmetric fractons, lineons and planons. In particular we have

$$\widetilde{Y}_{\mathcal{M}}^{(r)} = \prod_{e^{(g,y)}\in\mathcal{M}} Z_e \prod_{v\in\mathcal{M}}\prod_{e\ni v} X_e,\tag{52}$$

where $e^{(g,y)}$ denotes an edge shared by a green and yellow cube, see Fig. 7 (a), and similarly for other colors. One can also gauge $S_{\mathcal{M}}^{(c)}$ which results in a sum of operators each given by a product of $Z$ strings on edges connecting pairs of $Z_v$ terms in the ungauged operator times a product of $\prod_{e\ni v} X_e$ operators on the remaining vertices in $\mathcal{M}$.

The lineon string operators can be gauged in a similar fashion following Eq. (15)

$$\widetilde{Y}_{l_x}^{(r,b)} = \prod_{e^{(g,y)}\in l_x} Z_e \prod_{v\in l_x}\prod_{e\ni v} X_e,\tag{53}$$

and $S_{l_x}^{(r,b)}$ results in a sum of operators which can be found by following the same logic for gauging $S_{\mathcal{M}}^{(c)}$.

### 3.2.1 Relation between Checkerboard and X-Cube models

To understand the effect of gauging the Hadamard symmetry on the QSS sectors of the checkerboard model, we invoke a mapping between this model and two copies of the X-Cube model, as discussed in Ref. [94]. Specifically, it was shown that the checkerboard model is equivalent, up to finite depth local unitaries, to two copies of the X-Cube model:

$$UH_{CB}U^{\dagger} \equiv H_{\mathrm{triv.}}^{0} + H_{XC}^{1} + H_{XC}^{2},\tag{54}$$

where $H^0_{\text{triv.}}$ is a Hamiltonian in the trivial phase acting on some ancilliary degrees of freedom. The explicit form for the unitary $U$ can be found in Ref. [94].

The map between the two models changes the excitation basis as follows:

$$
\begin{aligned}
r1 &\mapsto l_y f & 1r &\mapsto f l_y \\
g1 &\mapsto T^{-1}_{\hat{y}+\hat{z}}(l_z f) & 1g &\mapsto f l_z \\
b1 &\mapsto T^{-1}_{\hat{x}+\hat{y}}(l_x f) & 1b &\mapsto f l_x \\
y1 &\mapsto T^{-1}_{\hat{x}+\hat{z}}(1f) & 1y &\mapsto f 1
\end{aligned}
\tag{55}
$$

where $T_{\hat{i}}$ is the translation operator along lattice direction $\hat{i} = \hat{x}, \hat{y}, \hat{z}$.

From this, we can see that the Hadamard symmetry of the checkerboard acts on the basis for two copies of the X-Cube as follows:

$$
\begin{aligned}
f1 &\leftrightarrow T_{\hat{x}+\hat{z}}(1f), \\
l_x 1 &\leftrightarrow T_{\hat{y}}(1l_x) \times T_{\hat{y}}(1f) \times T_{\hat{z}}(1f), \\
l_z 1 &\leftrightarrow T_{\hat{y}}(1l_z) \times T_{\hat{x}}(1f) \times T_{\hat{y}}(1f),
\end{aligned}
\tag{56}
$$

where we have used that $T_{\hat{i}}$ leaves an $l_i$ lineon sector invariant. While the Hadamard symmetry exchanges a fracton in the first X-Cube layer $f1$ with one in the second X-Cube layer $1f$, it does not act simply as layer exchange on the lineons. However, since $T_{\hat{y}}(1f) \times T_{\hat{z}}(1f)$ and $T_{\hat{x}}(1f) \times T_{\hat{y}}(1f)$ are both products of *planons*, the Hadamard symmetry indeed acts as layer exchange symmetry *on the QSS sectors* of the two X-Cube copies.

Thus, as far as the QSS sectors are concerned, through the explicit mapping given above we see that gauging the Hadamard symmetry of the checkerboard is equivalent to gauging the SWAP symmetry between two copies of the X-Cube model. Following our discussion of the latter, the excitation spectrum of the gauged Checkerboard model will also consist of fractons, lineons, and gauged flux loops, all of which are non-Abelian, as well as fully mobile Abelian particles. By analogy with the previous section, we can also establish the presence of *inextricably* non-Abelian fractons and lineons in the gauged model, as well as non-Abelian gauge flux loops.

We end our discussion of type-I models with an important open question: since the Hadamard symmetry acts as layer swap on the QSS sectors the X-Cube layers, it is natural to expect that the gauged checkerboard and gauged X-Cube layers will continue to be equivalent as foliated fracton phases [92]. However, demonstrating this first requires an extension of the notion of QSS sectors to non-Abelian fracton orders and then to panoptic fracton orders, since the presence of the genon loop excitations in the latter must also be accounted for. Moreover, since the additional degeneracy induced by the genon loops will also depend on the number of planons they can absorb, due to the non-trivial action of the Hadamard symmetry on the *bare* superselection sectors, it is conceivable that the gauged models are in distinct phases. We leave a detailed investigation of this question to future work.

## 3.3 Copies of Cubic Code with Layer Swap Symmetry

The Hamiltonian for two copies of the cubic code [2] is

$$
H_{2 \times CC} = -\sum_c \left( \mathcal{X}_c + \mathcal{Z}_c + \widetilde{\mathcal{X}}_c + \widetilde{\mathcal{Z}}_c \right),
\tag{57}
$$

where the Hamiltonian terms on the first layer are

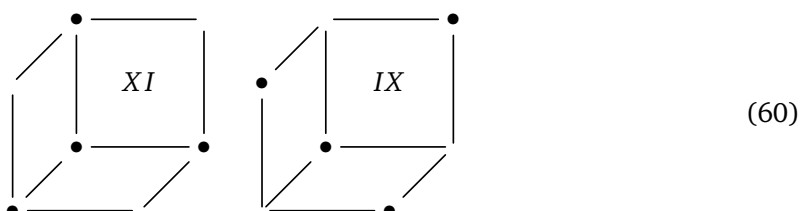

Similarly the terms acting on the second layer $\widetilde{\mathcal{X}}_c$, $\widetilde{\mathcal{Z}}_c$ are given by making the replacement $\left(X \to \widetilde{X}, Z \to \widetilde{Z}\right)$ in the above. The Hamiltonian respects the $\mathbb{Z}_2$ symmetry generated by swapping the qubits in each layer, with the following four qubit on-site action:

$$U_\nu = \text{SWAP} \otimes \text{SWAP}, \tag{58}$$

which acts as follows

$$XI \leftrightarrow \widetilde{X}I, \quad IX \leftrightarrow I\widetilde{X},$$
$$ZI \leftrightarrow \widetilde{Z}1, \quad IZ \leftrightarrow I\widetilde{Z}. \tag{59}$$

Excitations of $\mathcal{Z}_c$ in the first cubic code layer can be created in the following local clusters, shown on the dual cubic lattice,

$$\tag{60}$$

and similarly for excitations in the second layer by replacing $X \to \widetilde{X}$. The behavior of the $\mathcal{X}_c$ excitations can again be similarly obtained due to a combined spatial-parity + Hadamard + left-right qubit swap symmetry of the cubic code Hamiltonian. It was shown in Ref. [2] that there exist *no* string operators capable of moving topologically nontrivial excitations in the cubic code, hence making it a type-II model in the taxonomy of Ref. [8]. Single fracton excitations can be isolated by applying $XI$ or $IX$ to the sites in a discrete Sierpinski prism with the same orientation as the local charge clusters respectively [91, 95]. We denote the fractons in the bilayer system by $f_X 1, f_Z 1, 1f_X, 1f_Z$, suppressing their location on the dual lattice. The swap symmetry acts on the fractons in the obvious way.

Gauging the layer swap symmetry of $H_{2 \times CC}$, we find

$$H = -\sum_c \left(\mathcal{G}[\mathcal{X}_c + \widetilde{\mathcal{X}}_c] + \mathcal{G}[\mathcal{Z}_c + \widetilde{\mathcal{Z}}_c]\right) + H_{\mathcal{B}} + \Delta_P H_P, \tag{61}$$

for $H_{\mathcal{B}}$ and $H_P$ given in Eqs. (34) and (35).

As discussed in section 2.3, the gauged model supports non-Abelian fractons $[1f_{X/Z}]$, which are created at the corner of fractal operators since the gauging map preserves operator support. The gauged theory also contains Abelian fractons $[f_{X/Z}f_{X/Z}, \pm]$, which originate from a pair of fractons from both layers at a common coordinate. More generally, any $[f_{X/Z}f'_{X/Z}]$ particle with fractons $f_{X/Z}$ and $f'_{X/Z}$ appearing at distinct coordinates becomes non-Abelian. Interestingly, gauging the global swap symmetry results in a loss of the type-II no strings property as $\mathbb{Z}_2$ gauge charges $[11, \pm]$ with three dimensional mobility are introduced in the gauged

theory. Following our general discussion in Sec. 2, gauging also introduces non-Abelian gauge flux loops, whose quantum dimension depends on their size and shape as they can absorb any Abelian fracton $[f_{X/Z}f_{X/Z},+]$ that is incident on the loop. The number of such particles in distinct superselection sectors determines the quantum dimension, and is inherited from the properties of the superselection sectors in the cubic code. We leave a detailed quantitative analysis to future work.

The phase of matter obtained by gauging copies of the cubic code clearly lies beyond phases described by some decoupled conventional gauge theory and type-II fracton model, since the non-Abelian fractons in our model can absorb the gauge charges and have braid non-trivially with the gauge flux loops. Unlike the gauged foliated type-I models explained in the previous sections, where there is some precedent for a gapped order with fractons, mobile excitations, and loop excitations [63], to the best of our knowledge no model with similar properties to a gauged type-II model has appeared previously. In particular, the gauged cubic code represents the first example of a gapped phase hosting non-Abelian fractons at the corners of *fractal* operators. Note that the general arguments presented in Sec. 2.3 imply that the non-Abelian fractons are *inextricable*.

Similarly to more familiar non-Abelian anyons, e.g. the Ising $\sigma$ particle, a pair of defects with vacuum total charge can be used to encode a qubit in the gauge charge of an individual cluster. A logical operator pair is then given by a string operator moving a charge from one defect cluster to the other and a membrane braiding a loop excitation around a cluster. Unlike conventional non-Abelian anyons, if the defect clusters are sufficiently large their inherently slow dynamics [1, 3, 31, 32] should serve to keep them in place for long times without the need for pinning fields. It would be interesting to find other possible encodings with no string logical operators and hence larger code distances and energy barriers, and more favorable thermalization behavior for the encoded quantum information.

# 4 Discussion and conclusion

We have studied the effects of gauging a global symmetry that exchanges anyons between layers of fracton orders of type-I and type-II. Surprisingly, we find that this constructive approach leads to a new class of models with qualitatively novel behavior, since the resulting theories all host non-Abelian fractons alongside non-Abelian loop excitations and Abelian 3D particles. Besides unveiling a novel and distinct possiblity for gapped quantum phases in 3D, our work has potential implications for quantum computation, since the degenerate subspaces based on configurations of non-Abelian fractons could be useful for topological encoding of quantum information.

In essence, the *panoptic* fracton order we have found constitutes a hybrid order in which properties of fracton order and 3D TQFTs are non-trivially enmeshed: while retaining the geometric sensitivity and restricted mobility excitations of fracton phases, there appear additional excitations reminiscent of 3D TQFTs. This is similar in spirit to the string-membrane-net models introduced in Ref. [63]; since the construction here encompasses both non-Abelian fractons as well as gauged type-II models, we posit that the class of possible 3D phases is broader than that covered by the string-membrane-nets.

As the gauging map is invertible, fracton orders can clearly be derived from panoptic fracton states by "ungauging" the symmetry or condensing the gauge charges. Establishing whether there exists an analogous procedure through which the panoptic fracton state can be reduced to a pure TQFT[8] constitutes an important question going forwards, as an affirmative

---

[8]For conventional fracton orders, one can ungauge e.g., the planar sub-system symmetries of the X-Cube model, which reduces it to decoupled layers of 2D toric codes. A subsequent gauging procedure [14] then results in the

answer would show that panoptic fracton phases constitute a "parent" order, from which both TQFTs and fracton orders can descend. A natural candidate is a generalized condensation of some Abelian fracton particles to "ungauge" the fractal symmetry present in a type-II fracton order [13].

A similar line of inquiry is to better understand the generalized gauge theory that underlies the phases we have found. A straightforward first step in this direction would be finding *Abelian* panoptic orders, where Abelian subdimensional excitations co-exist with Abelian flux loops and fully mobile 3D particles. Considering such models could provide insights into whether they can be alternatively generated by gauging "hybrid" fractal/sub-system and $n$-form symmetries. Recall that gauging the swap symmetry of two copies of the quantum double $\mathcal{D}(G)$ leads to a model in the same phase as $\mathcal{D}((G \times G) \rtimes \mathbb{Z}_2)$; here, in some sense we have replaced the global $G$ symmetries with sub-system symmetries. Further analysis of our models could hence shed light on the nature of these new gauge theories and their appropriate algebraic structure, thereby providing a route to study generalizations of the quantum double description of ordinary discrete gauge theories. This would be of particular interest given that there currently exists no such general algebraic description even for usual fracton orders.

We have restricted our focus to gauging $\mathbb{Z}_2$ global symmetries here as this simple class already captures the conceptually novel features of interest. Our approach generalizes straightforwardly to any on-site, unitary representation of a finite group by following the lattice gauging procedure described in Sec. 2. Considering more general non-Abelian groups would lead to non-Abelian 3D particles, which are not of particular interest in the study of fractons as these are already captured within the familiar TQFT framework. For example, one could take $N > 2$ copies of a fractonic lattice model and gauge any subgroup of the $S_N$ global permutation symmetry group of such a model. Alternatively, we could also consider gauging the swap symmetry between copies of the X-cube model based on an arbitrary Abelian group $G$. More generally, understanding whether two foliated-equivalent phases remain equivalent as foliated fracton phases *after* gauging their appropriate on-site symmetries remains an important direction for future research, one which we believe can be understood within the general framework described here.

Our study of global symmetry actions on fracton orders also points to many interesting future directions on the way to a comprehensive theory of *symmetry-enriched* fracton order. One interesting aspect we plan to study in a forthcoming work is the fractionalization of a global symmetry on fractonic particles. The study of global symmetries is also interesting from a quantum codes perspective, specifically the question of what transversal or locality preserving gates are possible in codes based on fracton lattice models. Also, since gauging has been studied numerically and analytically within the Projective Entangled Pair States (PEPS) [84,85,96] formalism in terms of an internal/virtual gauge symmetry group, it would be interesting to further our understanding of such a symmetry group for fractonic tensor networks [54]. As a final remark, we note that while the gauging procedure has also been carried out for topological orders at the level of their low-energy effective field theories [97,98], whether it extends to the tensor gauge theory formalism remains an open question. Ideas in this direction may offer a route towards realizing non-Abelian symmetric tensor gauge theories, which have thus far proven elusive.

# Acknowledgements

We thank Dave Aasen, Meng Cheng, Michael Hermele, Albert Schmitz, and Yizhi You for stimulating discussions. A. P. acknowledges support through a PCTS fellowship at Princeton Uni-

---

3D toric code.

versity.

**Note:** During the course of this project we became aware of similar results, obtained independently in Ref. [99] by D. Bulmash and M. Barkeshli, which appeared in the same arXiv posting.

# A Gauging layer swap symmetry of two copies of toric code

## A.1 2D toric code

We consider two copies of the toric code [73] on the square lattice to illustrate the gauging procedure on a familiar lattice model. We place two qubits on each link of the lattice, with Pauli operators $X, Z$ corresponding to the first qubit and $\widetilde{X}, \widetilde{Z}$ to the second. In order to make the gauging procedure transparent, we first shift the qubits from the links to the sites of the lattice, as depicted in Fig. 8 for a single toric code copy. Upon regrouping, there are four qubits on each site, with Hamiltonian

$$H_{2\times 2DTC} = -\sum_p \left( \mathcal{X}_p + \mathcal{Z}_p + \widetilde{\mathcal{X}}_p + \widetilde{\mathcal{X}}_p \right), \tag{62}$$

where the plaquette (p) terms $\mathcal{X}_p$ and $\mathcal{Z}_p$ are given by

$$\mathcal{X}_p = \begin{matrix} XI - XX \\ | \quad | \\ II - IX \end{matrix} \quad , \quad \mathcal{Z}_p = \begin{matrix} ZI - II \\ | \quad | \\ ZZ - IZ \end{matrix} \quad , \tag{63}$$

respectively, which act on the first toric code layer. The terms acting on the second layer $\widetilde{\mathcal{X}}_p, \widetilde{\mathcal{Z}}_p$ are given by making the replacement $\left( X \to \widetilde{X}, Z \to \widetilde{Z} \right)$.

The above Hamiltonian respects the on-site $\mathbb{Z}_2$ symmetry generated by exchanging the layers *i.e.,* swapping the qubits in each layer, with the four qubit on-site action: $U_{x,y} = \text{SWAP} \otimes \text{SWAP}$, which acts as follows on the four qubits residing on site $(x, y)$:

$$X_{x,y}^{\hat{i}} \longleftrightarrow \widetilde{X}_{x,y}^{\hat{i}}, \quad Z_{x,y}^{\hat{i}} \longleftrightarrow \widetilde{Z}_{x,y}^{\hat{i}}. \tag{64}$$

Following the general gauging procedure delineated in Sec. 2, we introduce an additional qubit on each edge $(x + 1/2, y), (x, y + 1/2)$ and modify the Hamiltonian accordingly:

$$H = -\sum_p \mathcal{G}\left[ \mathcal{X}_p + \widetilde{\mathcal{X}}_p \right] - \sum_c \mathcal{G}\left[ \mathcal{Z}_p + \widetilde{\mathcal{Z}}_p \right] + H_{\mathcal{B}}^{2D} + H_P^{2D}, \tag{65}$$

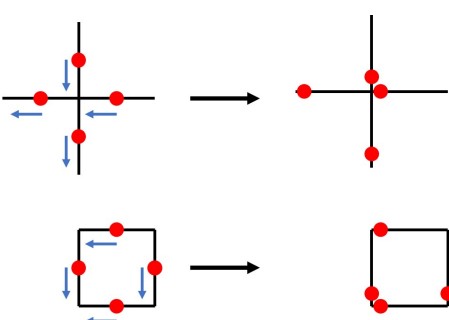

Figure 8: Qubits originally living on the links of a square lattice are regrouped on to vertices as shown here.

where the first term in the above Hamiltonian acts on gauge qubits $(x - 1/2, y), (x, y - 1/2)$, while the next acts on $(x + 1/2, y), (x, y + 1/2)$, and

$$H_{\mathcal{B}}^{2D} = -\sum_{x,y} Z_{x+\frac{1}{2},y} Z_{x,y+\frac{1}{2}} Z_{x+\frac{1}{2},y+1} Z_{x+1,y+\frac{1}{2}}, \tag{66}$$

$$H_{P}^{2D} = -\sum_{x,y} U_{x,y} X_{x+\frac{1}{2},y} X_{x,y+\frac{1}{2}} X_{x-\frac{1}{2},y} X_{x,y-\frac{1}{2}}. \tag{67}$$

The swap symmetry acts as follows on the generating anyons:

$$e1 \leftrightarrow 1e, \quad m1 \leftrightarrow 1m. \tag{68}$$

Hence anyons of the form $aa$ are fixed by the symmetry, while anyons of the form $ab$, with $a \neq b$, form cycles of length two.

The symmetry defects are genons which appear at the end of layer swap domain walls (see Fig. 1). These defects can carry a charge under each string operator associated to a symmetric anyon. Hence, there are four genons $g_{\pm\pm}$, labelled by their eigenvalues under braiding with $ee, mm$, respectively. As these genons are all related by fusion with Abelian anyons of the form $a1$, they consequently all have the same quantum dimension. The genons are non-Abelian since they can absorb anyons of the form $aa$, which can be split into $a1 \otimes 1a$ and moved around the genon to annihilate. The total quantum dimension of the defects matches that of the anyons, hence the quantum dimension of each genon is $d_{g_{\pm\pm}} = 2$.

Upon gauging the layer swap symmetry, the anyons of the form $aa$ split into 8 different Abelian anyons labelled by $[aa, \pm]$. Pairs of anyons of the form $ab, ba, a \neq b$, coalesce into single non-Abelian anyons $[ab]$ with quantum dimension $d = 2$. There are 6 such anyons in total. Finally, the genons split into 8 anyons with quantum dimension 2 labelled by $[g_{\pm\pm}, \pm]$, so that the total number of superselection sectors in the gauged theory is 22. The resulting topological order is equivalent to the discrete gauge theory based on $(\mathbb{Z}_2 \times \mathbb{Z}_2) \rtimes \mathbb{Z}_2 \cong D_4$, which we denote by $\mathcal{D}(D_4)$.

## A.2 3D toric code

Similarly to the previous section, we consider two copies of the 3D toric code on the cubic lattice, where we group three qubits (per layer) onto each site. Upon regrouping, the Hamiltonian is given by

$$\begin{aligned} H_{2\times TC} = &-\sum_c \left( \mathcal{X}_c + \mathcal{Z}_c^x + \mathcal{Z}_c^y + \mathcal{Z}_c^z \right) \\ &-\sum_c \left( \widetilde{\mathcal{X}}_c + \widetilde{\mathcal{Z}}_c^x + \widetilde{\mathcal{Z}}_c^y + \widetilde{\mathcal{Z}}_c^z \right), \end{aligned} \tag{69}$$

where the usual vertex and plaquette terms of the 3D toric code are now represented as the following cube terms:

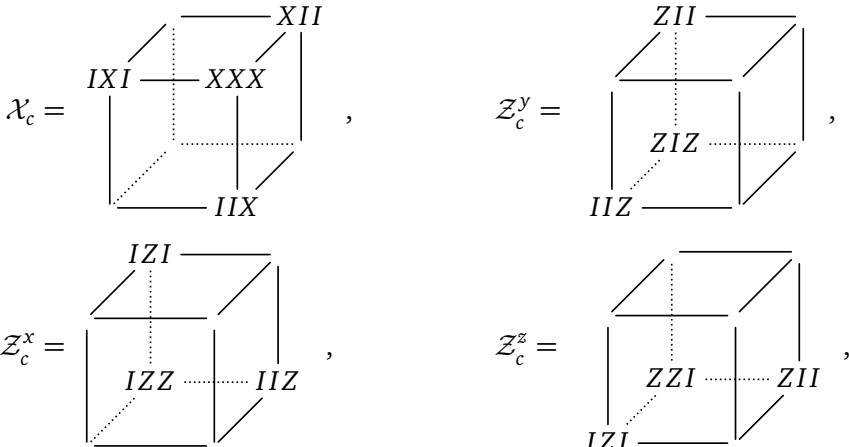

which all act on the first toric code layer. The terms acting on the second layer are obtained by taking $\left(X \to \widetilde{X}, Z \to \widetilde{Z}\right)$ in the above.

The Hamiltonian written above respects the on-site $\mathbb{Z}_2$ symmetry generated by

$$U_v = \text{SWAP} \otimes \text{SWAP} \otimes \text{SWAP}, \tag{70}$$

which acts as follows on the qubits located at each vertex $v = (x, y, z)$

$$X^{\hat{i}}_{x,y,z} \longleftrightarrow \widetilde{X}^{\hat{i}}_{x,y,z}, \qquad\qquad Z^{\hat{i}}_{x,y,z} \longleftrightarrow \widetilde{Z}^{\hat{i}}_{x,y,z}. \tag{71}$$

To gauge the symmetry we again introduce a qubit on each edge $(x + 1/2, y, z), (x, y + 1/2, z), (x, y, z + 1/2)$ so that the Hamiltonian becomes

$$H = -\sum_c \mathcal{G}\big[\mathcal{X}_c + \widetilde{\mathcal{X}}_c\big] - \sum_c \mathcal{G}\big[\mathcal{Z}^x_c + \widetilde{\mathcal{Z}}^x_c\big] - \sum_c \mathcal{G}\big[\mathcal{Z}^y_c + \widetilde{\mathcal{Z}}^y_c\big] - \sum_c \mathcal{G}\big[\mathcal{Z}^z_c + \widetilde{\mathcal{Z}}^z_c\big]$$
$$+ H_{\mathcal{B}} + \Delta_P H_P, \tag{72}$$

where the first term acts on gauge qubits $(x-1/2, y, z), (x, y-1/2, z), (x, y, z-1/2)$, the second term acts on $(x+1/2, y, z), (x, y+1/2, z)$, and the third and fourth terms act analogously. The term $H_{\mathcal{B}}$ is defined in Eq. (34), and $H_P$ is defined in Eq. (35).

The swap symmetry acts as follows on the generating point and looplike excitations

$$e1 \longleftrightarrow 1e, \quad m^\gamma 1 \longleftrightarrow 1m^\gamma. \tag{73}$$

Hence $ee$ is fixed under the symmetry, and $1e, e1$ form a cycle of length 2. The looplike excitations behave similarly.

The symmetry defects are looplike genons appearing at the boundary of layer swap domain walls. These defects come in two types: $g^\gamma_\pm$, distinguished by their eigenvalue under a linking string operator generated by $ee$. The two defects are related by fusion with $m^\gamma 1$. The genon loops are non-Abelian, and carry a cheshire charge, as they can absorb $ee$ particles.

Gauging the layer swap symmetry causes the vacuum and $ee$ particle to split into pairs of Abelian charges $[11, \pm]$ and $[ee, \pm]$, respectively. Meanwhile $e1$ and $1e$ coalesce into a single non-Abelian particle $[e1]$ with quantum dimension 2. Similarly, $m^\gamma 1$ and $1m^\gamma$ coalesce into a single loop excitation $[m1]^\gamma$ which is non-Abelian, and carries cheshire charge as it can absorb $[11, -]$ particles. On the other hand $m^\gamma m^\gamma$ also leads to a single loop excitation $[mm]^\gamma$ which is Abelian. The genon loops again lead to single loop excitations $[g_\pm]^\gamma$ which remain non-Abelian, and can absorb $[ee, +]$ and $[ee, -]$, respectively. The resulting topological order is described by $D_4$ gauge theory.

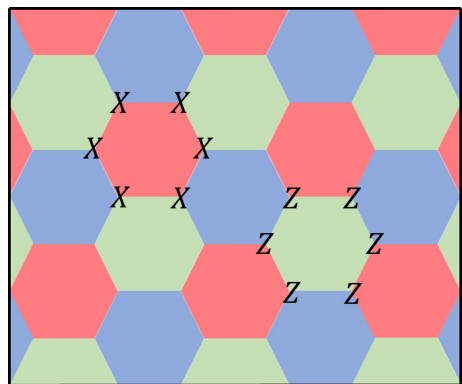

Figure 9: The 2D color code is defined on the honeycomb lattice with one physical qubit per vertex. For each plaquette, there are two stabilizers given by the product of either $X$ or $Z$ operators on the vertices forming the plaquette.

## B  Gauging the Hadamard symmetry of the 2D color code

The 2D color code [100] is defined on the honeycomb lattice, with one qubit per vertex, with the Hamiltonian given by

$$H = -\sum_p \prod_{v \in \partial p} X_v - \sum_p \prod_{v \in \partial p} Z_v. \tag{74}$$

It is convenient to pick a three coloring $r, g, b$ of the honeycomb lattice, as shown in Fig. 9, to describe the excitations of this model. Correspondingly, we label excitations according to the color of the plaquette stabilizer they violate and by the stabilizer type—$X$ or $Z$—they violate. Specifically, we label excitations by the ordered pair $c_X c_Z$ where $c_X (c_Z) = r, g, b$ indicates the color of the $X$-type ($Z$-type) stabilizer violated. For example, $rb$ denotes an excitation of a red $X$ stabilizer and blue $Z$ stabilizer.

An (over-complete) generating set of string operators for the sixteen anyonic excitations of the color code is given by

$$X^{(c)}_{\langle p,q \rangle} = \prod_{v \in \langle p,q \rangle} X_v, \quad Z^{(c)}_{\langle p,q \rangle} = \prod_{v \in \langle p,q \rangle} Z_v, \tag{75}$$

where $\langle p, q \rangle$ is a path between plaquettes $p, q$ of the same color $c = r, g, b$, along disjoint edges connecting pairs of $c$-plaquettes. As can be checked explicitly, excitations obey the fusion rules

$$c_X 1 \times c_X 1 = 1, \quad 1 c_Z \times 1 c_Z = 1, \quad (\forall\, c_X, c_Z = r, g, b), \tag{76}$$

$$x_1 1 \times x_2 1 = x_3 1, \quad 1 z_1 \times 1 z_2 = 1 z_3, \tag{77}$$

for pair-wise distinct color labels $x_1, x_2, x_3$ and $z_1, z_2, z_3$. Thus, {r1, g1, 1r, 1g} is a minimal generating set for the color code anyons.

Note that the total number of excitations of the color code is equivalent to that of two copies of the 2D toric code, as is the number of minimal generating anyons. The 2D color code is in fact exactly equivalent, under a finite-depth local unitary circuit, to two copies of the 2D toric code [88–90]. In terms of anyonic excitations, the mapping to copies of toric code is given through the correspondence:

$$
\begin{aligned}
r1 &\mapsto e1, & g1 &\mapsto 1m, & b1 &\mapsto em, \\
1r &\mapsto 1e, & 1g &\mapsto m1, & 1b &\mapsto me.
\end{aligned}
\tag{78}
$$

Now, the on-site $\mathbb{Z}_2$ symmetry of the color code is generated by $h^{\otimes N}$ where $h$ is the Hadamard matrix defined previously in Eq. (40). The symmetry acts on the excitations in either basis as follows

$$ab \leftrightarrow ba\,, \tag{79}$$

clearly acting as layer swap on the copies of toric code. Therefore, the emergent symmetry-enriched topological order, as well as the outcome of gauging, is identical to that of the previous section A.1. The only distinction is the precise Hamiltonian, and representation, being considered.

To gauge the symmetry it is convenient to change basis: $h \mapsto X$, so that, up to an overall normalization factor, the Hamiltonian (74) becomes

$$H = -\frac{1}{2}\sum_p \prod_{v\in\partial p}(X_v + Z_v) - \frac{1}{2}\sum_p \prod_{v\in\partial p}(X_v - Z_v) = -\sum_p \sum_{i=0}^{3} B_p^{(2i)}\,, \tag{80}$$

where

$$B_p^{(j)} := \sum_{s\in\mathbb{Z}_2^6,\,\mathrm{wt}(s)=j} \prod_{v=1}^{6} X_v^{1+s_v} Z_v^{s_v}\,. \tag{81}$$

In the above equation $v = 1,\ldots 6$, label the vertices of plaquette $p$, and $\mathrm{wt}(s)$ is the weight function equal to the number of nonzero entries in the vector $s$.

The symmetric string operators are given by

$$
\begin{aligned}
Y_{\langle p,q\rangle}^{(c)} &= \prod_{v\in\langle p,q\rangle} Y_v\,, \\
S_{\langle p,q\rangle}^{(c)} &= \frac{1}{2}\prod_{v\in\langle p,q\rangle}(X_v + Z_v) + \frac{1}{2}\prod_{v\in\langle p,q\rangle}(X_v - Z_v)\,,
\end{aligned} \tag{82}
$$

for the $cc$ and $c1+1c$ anyons, respectively, where $c = r, g, b$. The operator $S_{\langle p,q\rangle}^{(c)}$ is given by a sum over all products of either $X_v$ or $Z_v$ on each vertex in $\langle p,q\rangle$, with the constraint that the number of $Z_v$ operators in the string must be even.

Since the symmetry acts on-site as the regular representation, it is simple to carry through the gauging and disentangling steps in one go following the recipe in Eq. (15). This switches the variables to qubits on the edges of the honeycomb lattice, with Hamiltonian

$$H = -\sum_p \sum_{i=0}^{3} \widetilde{B}_p^{(2i)} + \prod_{e\in\partial p} Z_e\,. \tag{83}$$

In the above equation

$$\widetilde{B}_p^{(j)} := \sum_{s\in\mathbb{Z}_2^6,\,\mathrm{wt}(s)=j} F_p^{(s)} \prod_{v=1}^{6} A_v^{1+s_v}\,, \tag{84}$$

where

$$A_v = \prod_{e\ni v} X_e\,, \tag{85}$$

$$F_p^{(s)} = \prod_{e=1}^{6} Z_e^{\tilde{s}_e}\,, \tag{86}$$

for $e = 1, \ldots, 6$, the edges of plaquette $p$ and

$$\tilde{s}_e = \sum_{i=1}^{e} s_i \mod 2 \,. \tag{87}$$

We can also gauge the string operators for symmetric anyons following Eq. (15)

$$\widetilde{Y}_{\langle p,q \rangle}^{(c)} = \prod_{e \in \langle p,q \rangle} Z_e \prod_{e' \in N_e} X_{e'} \,, \tag{88}$$

where $N_e$ denotes edges sharing a vertex with $e$. Gauging the string operator $S_{\langle p,q \rangle}^{(c)}$ can be done similarly, leading to a sum of operators with $Z$ strings along edges between pairs of $Z_v$ in the ungauged operator, and replacing each $X_v$ with a star term.

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
