# Peer review of "Gauging permutation symmetries as a route to non-Abelian fractons"

_SciPost Physics, doi:SciPost Phys. 7, 068 (2019)_

## Round 2 · Referee Report · Anonymous · 2019-8-5

Report

This paper presents the explicit construction of exactly solvable fracton models with novel features. In particular, the models are obtained by gauging a Z2 permutation symmetry in known fracton models and they exhibit new features like the co-existence of the topological type excitations and fracton type excitations, inextricably non-abelian fractons, etc. These features were not present in the models that were previously known, therefore the new models expand our knowledge about fracton models and the paper presents an important contribution to the field.

The paper is written in an amazingly clear way. Pedagogical examples were explained in detail. Moreover, a systematic discussion was given to explain the appearance of the new features under a generic setup. The paper should be easy to follow for people working in the field. Therefore, I recommend publication of this paper pretty much as it is. I only have one minor comment: The caption of figure 2 is a bit confusing. The figure contains (a) and (b) parts which are not referred to in the caption. How is the blue region different from the green region in figure (b)?

---

## Round 3 · Author Response

Dear Editor and Referee,

Thank for your careful consideration and review of our work. We thank the referee for their kind remarks regarding our manuscript. In accordance with the referee's report, we have clarified the caption for Fig. 2, where the two colours (blue/green) are used to distinguish between the distinct layers.

Sincerely,
Abhinav Prem and Dominic Williamson

---

## Round 3 · List of Changes

1) We have clarified the caption for Figure 2.

---

## Editorial Decision

published